# Ternary nickel–tungsten–copper alloy rivals platinum for catalyzing alkaline hydrogen oxidation

Shuai Qin[1,5], Yu Duan[1,5], Xiao-Long Zhang [1,5], Li-Rong Zheng[2], Fei-Yue Gao[1], Peng-Peng Yang[1], Zhuang-Zhuang Niu[1], Ren Liu[3], Yu Yang[1], Xu-Sheng Zheng[4], Jun-Fa Zhu [4] & Min-Rui Gao [1✉]

Operating fuel cells in alkaline environments permits the use of platinum-group-metal-free (PGM-free) catalysts and inexpensive bipolar plates, leading to significant cost reduction. Of the PGM-free catalysts explored, however, only a few nickel-based materials are active for catalyzing the hydrogen oxidation reaction (HOR) in alkali; moreover, these catalysts deactivate rapidly at high anode potentials owing to nickel hydroxide formation. Here we describe that a nickel–tungsten–copper ($Ni_{5.2}WCu_{2.2}$) ternary alloy showing HOR activity rivals Pt/C benchmark in alkaline electrolyte. Importantly, we achieved a high anode potential up to 0.3 V versus reversible hydrogen electrode on this catalyst with good operational stability over 20 h. The catalyst also displays excellent CO-tolerant ability that Pt/C catalyst lacks. Experimental and theoretical studies uncover that nickel, tungsten, and copper play in synergy to create a favorable alloying surface for optimized hydrogen and hydroxyl bindings, as well as for the improved oxidation resistance, which result in the HOR enhancement.

[1] Division of Nanomaterials & Chemistry, Hefei National Laboratory for Physical Sciences at the Microscale, University of Science and Technology of China, Hefei, China. [2] Beijing Synchrotron Radiation Facility, Institute of High Energy Physics, Chinese Academy of Sciences, Beijing, China. [3] Department of Materials Science and Engineering, University of Science and Technology of China, Hefei, China. [4] National Synchrotron Radiation Laboratory, University of Science and Technology of China, Hefei, China. [5] These authors contributed equally: Shuai Qin, Yu Duan, Xiao-Long Zhang. ✉email: mgao@ustc.edu.cn

Over the past 30 years, the proton exchange membrane fuel cell (PEMFC) technology has developed rapidly, resulting in the first commercial sales of fuel-cell powered cars in 2015 (ref. [1]). Although great success, the mass market penetration by such zero-emission vehicles is currently hindered by the absence of inexpensive materials to replace costly platinum (Pt)-based catalysts, which are responsible for ~46% of the stack cost[2]. Extensive research has been conducted using various strategies to reduce the Pt loading[3–5], but the acidic environment largely constrains the choice of catalytic materials that perform stably in PEMFCs. This shortcoming can be overcome by switching the operating environment from an acid to alkaline one-the resultant anion exchange membrane fuel cells (AEMFCs) allow for the use of platinum group metal-free (PGM-free) catalysts, thus enabling significant cost reduction[6]. Indeed, some PGM-free materials have been observed to catalyze the oxygen reduction reaction (ORR) comparable to Pt for the AEMFC cathode[7–10]. However, at the AEMFC anode, the activity of the hydrogen oxidation reaction (HOR) on Pt is inherently slower by ~100 times than that in acidic environment[11,12], which demands much higher Pt loadings (0.4 $mg_{Pt}$ $cm^{-2}$ at AEMFC anode versus 0.03 $mg_{Pt}$ $cm^{-2}$ at PEMFC anode) to reach similar fuel cell performance[13]. At present, the lack of highly active and stable PGM-free HOR catalysts in alkaline environments hampers the progress towards the AEMFC implementation[14].

Substantial effort has been devoted to searching HOR catalysts composed solely of earth-abundant elements, whereas PGM-free catalysts that show HOR activity in alkali are rather rare[15]. Nickel (Ni)-based compounds are currently the materials with the most promise as catalysts that drive the HOR in alkaline environments[16]. In 1960s, Raney Ni was first explored as HOR catalyst in liquid alkaline fuel cells under extreme alkalinity (6M KOH), but its activity was very low[17–19]. Since then, the HOR performance of Ni-based catalysts has been gradually improved via diverse methods, yielding catalysts such as CoNiMo[20], MoNi$_4$ and WNi$_4$[21], NiMo/C[22], Ni/NiO/C[23], Ni/N-doped carbon nanotubes[13], CeO$_2$/Ni heterostructures[24], Cr-modified Ni[25], and Ni$_3$N nanoparticles[26,27]. Despite marked progress, many of these Ni-based HOR catalysts deactivate quickly above 0.1 V versus reversible hydrogen electrode (RHE) owing to the formation of surface hydroxides[28]. To generate adequate power density, however, HOR catalysts should remain stable to at least 0.3 V versus RHE[2], which could also largely mitigate the risk of passivation of the stack under transient conditions. Unfortunately, there is no PGM-free HOR catalyst with operating stability up to 0.3 V versus RHE has been reported thus far.

Here we report that Ni$_{5.2}$WCu$_{2.2}$ nanotubes (NTs)—a ternary nickel–tungsten–copper alloy—can catalyze the HOR in alkaline electrolyte highly efficient and stable. We find that the Ni$_{5.2}$WCu$_{2.2}$ catalyst achieves higher HOR activity than a commercial Pt/C counterpart. Importantly, stability window up to 0.3 V versus RHE was realized on this catalyst. At this potential, a considerable current density of 16.5 mA cm$^{-2}$ could be held over 20 hours without notable decay. Moreover, this alloyed catalyst also exhibits high tolerance to 20,000 ppm CO impurity. The extraordinary efficiency of Ni$_{5.2}$WCu$_{2.2}$ exceeds that of previously reported PGM-free HOR catalysts.

## Results

### Synthesis and characterization of catalyst.
Very recently, our group has reported bimetallic MoNi$_4$ and WNi$_4$ nanoalloys with notable activity toward the alkaline HOR, but such binary alloys lose activity above 0.2 V versus RHE owing to the surface oxidation[21]. Previous studies suggested an alloying-based approach for improved stability of catalysts by tuning the

microstructures of the host metals[29–32]. For examples, alloying Mo with Pt$_3$Ni (111) can mitigate the Ni leaching during electrochemical process, enabling greatly enhanced ORR stability[31]. Chorkendorff et al. identified that alloys of Pt and rare earths are stable ORR catalysts because of their very negative alloying energy $E_a$ that limits dissolution[32]. We thus seek to widen the stability window of binary Ni-W catalyst through alloying with additional element to form new structure that resists surface oxidation during HOR operation. Prior studies have shown that HOR activity and stability can be improved largely in alkaline electrolytes when using Cu to partially replace noble metals (e.g., PdCu nanoparticles[33] and Pt/Cu nanowires[34]). Moreover, Dekel et al. reported that Cu in alloys often dissolves at higher anode potentials as compared to other metals such as Mo[28]. We thus speculate that Cu may play an important role in enhancing the HOR activity and stability. Hence, we decide to introduce Cu to form ternary Ni-W-Cu alloy and explore its alkaline HOR property.

To synthesize the catalyst, Cu foam was immersed in 1M KOH for an anodization process to grow Cu(OH)$_2$ nanowires throughout the substrate (Supplementary Figs. 1 and 2). The resulting indigo foam was consequently treated hydrothermally with Ni(NO$_3$)$_2$·6H$_2$O, (NH$_4$)$_6$H$_2$W$_{12}$O$_{40}$·xH$_2$O, and CO(NH$_2$)$_2$ in deionized water at 130 °C for 8 h. Then, the obtained green NiW-Cu(OH)$_2$ precursor (Supplementary Figs. 3 and 4) was annealed in hydrogen/argon (H$_2$/Ar: 5/95) atmosphere at 500 °C for 1 h to yield ternary Ni-W-Cu alloy (Fig. 1a). This simple synthetic approach provides good scalability. We synthesized a 3 cm × 10 cm Ni-W-Cu alloy foam by using scaled-up reactors, which shows good fidelity of the product, implying a potential industry-level use (Fig. 1b and Supplementary Fig. 5).

The achieved electrode consists of numerous uniform Ni-W-Cu alloy nanofibers with length up to 10 μm when imaged by scanning electron microscopy (SEM) (Fig. 2a and Supplementary Fig. 6). Each fiber possesses a very rough surface that is made up of whiskerette particles. Low-resolution transmission electron microscopy (TEM) unveils that the center of a representative fiber appears brighter, demonstrating that it is hollow (Fig. 2b). The formation of hollow-fibers was caused by the outward diffusion of Cu in the NiW-Cu(OH)$_2$ precursor to produce Ni-W-Cu alloy during annealing process. Selected-area electron diffraction (SAED) analysis of a whisker reveals its high-crystalline structure (Up inset in Fig. 2b), which agrees with our atomic-resolution high-angle annular dark-field scanning transmission electron microscopy (HAADF-STEM) result that shows continuous lattice fringes without surface defects (Down inset in Fig. 2b). In Fig. 2c, the STEM measurements reveal that alloyed whiskerette particles bounded closely to form distinct grain boundaries. Energy-dispersive X-ray (EDX) spectrum elemental mapping displays a uniform, uncorrelated spatial distribution of Ni, W, and Cu (Fig. 2c and Supplementary Fig. 7). The weak O signal could originate from the adsorbed O and slight surface oxidation when exposing the sample in the air (Supplementary Fig. 8).

We performed X-ray diffraction (XRD) measurements of the collected Ni-W-Cu alloy powder and observed a high-crystallinity face-centred cubic structure (Fig. 2d). Compared to (111), (200), and (220) reflections of Ni nanoparticles, the diffraction peaks are shifted to lower angles in Ni-W-Cu alloy, indicating a higher lattice parameter caused by the addition of atoms with larger radius. Using similar synthetic method, we also prepared Ni-W alloy with cubic Ni$_{17}$W$_3$ phase (Fig. 2d and Supplementary Fig. 9), whose diffraction peaks sit between those of Ni nanoparticles (Fig. 2d and Supplementary Fig. 10) and Ni-W-Cu alloy. Given that the atomic radius follows the sequence of Ni < Cu < W, we predict more W atoms in Ni-W-Cu than in Ni$_{17}$W$_3$. Our inductively coupled plasma atomic emission

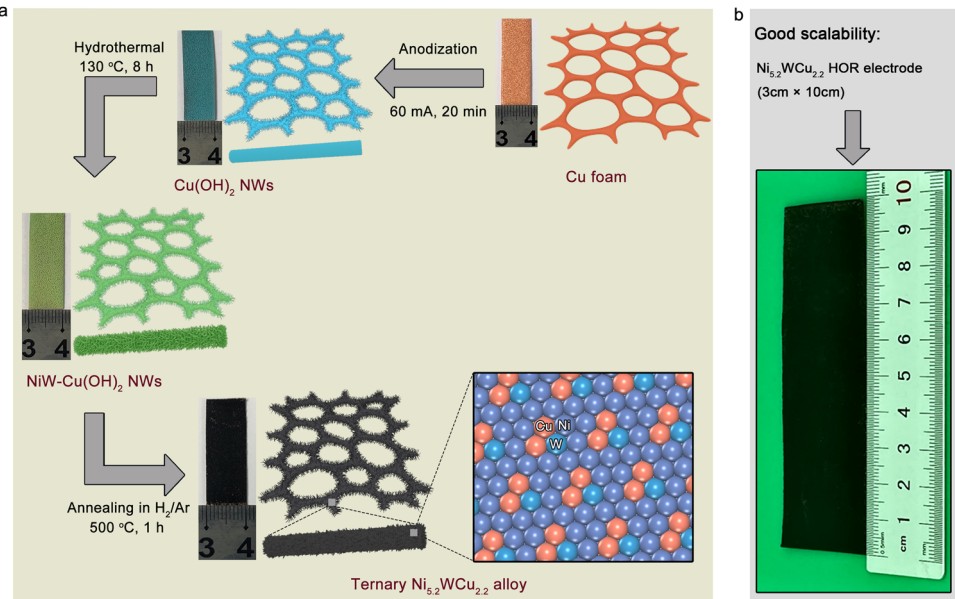

**Fig. 1 Synthesis of ternary Ni$_{5.2}$WCu$_{2.2}$ alloy. a** Schematic illustration of the synthesis of Ni$_{5.2}$WCu$_{2.2}$ alloy monolith. **b** Scaled-up synthesis of Ni$_{5.2}$WCu$_{2.2}$ alloy monolith with a size of 3 cm × 10 cm using the identical protocol.

spectroscopy (ICP-AES) analysis gives a Ni:W:Cu ratio of 5.2:1:2.2 for Ni-W-Cu alloy, which agrees well with EDX measurements and confirms our speculation. Thereafter, we assign the obtained ternary alloy as Ni$_{5.2}$WCu$_{2.2}$.

Alloying-based design has been demonstrated to alter catalytic properties by modulating the electronic structures of host and alloyed metals, leading to a cooperative interplay that tunes the adsorption energies of critical intermediates for enhanced catalytic reactivity[35–37]. We probed the work function of Ni$_{5.2}$WCu$_{2.2}$, Ni$_{17}$W$_3$ and Ni catalysts by Kelvin probe force microscopy[38,39] (KPFM; Supplementary Fig. 11). A clean highly oriented pyrolytic graphite (HOPG) with work function value of 4.6 eV was used as a reference[39]. Figure 2e presents the measured surface potentials for the studied samples, which correspond to a work function value of 4.54 for Ni$_{5.2}$WCu$_{2.2}$, versus 4.71 eV for Ni$_{17}$W$_3$ and 4.80 eV for Ni (Inset in Fig. 2e), consistent with our calculated results (Supplementary Fig. 12). The lower work function of Ni$_{5.2}$WCu$_{2.2}$ suggests its modified electronic structure that allows for a faster electron transfer and thus enhanced catalytic ability[40]. To reveal the impact of ternary alloy effect on the chemical and structural environments of Ni, we performed the X-ray absorption spectroscopy. Figure 2f shows the X-ray absorption near-edge spectroscopy measurements of the Ni K-edge from Ni$_{5.2}$WCu$_{2.2}$, Ni$_{17}$W$_3$, and freshly-prepared Ni samples, as well as that from a Ni foil used for comparison. Adsorption features that are present in the spectra of our freshly-prepared Ni and Ni foil reference are similar. By contrast, a substantial increase in the white lines was observed for Ni$_{5.2}$WCu$_{2.2}$ and Ni$_{17}$W$_3$, indicating electron donation from Ni to W and Cu upon alloying[41]. We probed the radial structure function around Ni by Fourier transform (FT) of extended X-ray absorption fine-structure (EXAFS) spectra (Fig. 2g, Supplementary Fig. 13). A profound peak at ~2.1 Å can be attributable to the Ni-Ni and Ni-W/Cu bonds in Ni$_{5.2}$WCu$_{2.2}$ and Ni$_{17}$W$_3$ alloys[21,42]. The intensity of this peak decreases upon alloying Ni with W and Cu to form Ni$_{5.2}$WCu$_{2.2}$, which represents damped coordination structure of Ni. Furthermore, our Ni K-edge EXAFS fittings reveal that the first-shell Ni-W/Cu coordination number decreases from freshly-prepared Ni (~10.8) to Ni$_{17}$W$_3$ (~9.0) and then to Ni$_{5.2}$WCu$_{2.2}$ (~8.7) (Fig. 2h,

Supplementary Fig. 14, Supplementary Table 1). We surmise the lower coordination number attributable to the highly nanostructured hollow structure of Ni$_{5.2}$WCu$_{2.2}$, and estimate that more active sites exist on the ternary alloy that would modify the adsorption ability of HOR intermediates.

**Evaluation of HOR performance.** We evaluated HOR activity and stability of the ternary Ni$_{5.2}$WCu$_{2.2}$ alloy in a standard three-electrode setup (Supplementary Fig. 15), using H$_2$-saturated 0.1 M aqueous KOH as the electrolyte. The working electrode cuts from large-area Ni$_{5.2}$WCu$_{2.2}$ monolith with a geometric surface area of ~1 cm$^2$ (catalyst loading: ~9.2 mg cm$^{-2}$). Reference measurements of Ni$_{17}$W$_3$, Ni, and commercial Pt/C (20 wt%) catalysts were similarly performed for comparison. We used a slow sweep rate of 1 mV s$^{-1}$ to minimize the capacitive contribution and to guarantee a steady-state measurement. Unless otherwise stated, all electrochemical data were measured without electrode rotation and were iR-corrected (i, current; R, resistance) for the uncompensated Ohmic drop.

As shown by the linear sweep voltammetry (LSV) curve of Ni$_{5.2}$WCu$_{2.2}$ in 0.1 M KOH, the onset potential for generating HOR current is as low as 0 V versus RHE, beyond which a sharp increase in anodic current was observed, demonstrating its exceptional energetics for HOR (Fig. 3a). By contrast, Ni$_{5.2}$WCu$_{2.2}$ gives almost no current-voltage feature in 0.1 M KOH saturated by Ar (Supplementary Fig. 16), further confirming that HOR catalysis occurs on this alloy catalyst. Figure 3a also shows that Ni$_{5.2}$WCu$_{2.2}$ can reach a diffusion-limiting HOR current density of ~20 mA cm$^{-2}$, whereas Ni$_{17}$W$_3$ exhibits much inferior HOR activity and the single Ni is almost HOR inactive. Remarkably, the Ni$_{5.2}$WCu$_{2.2}$ catalyst even surpasses the commercial Pt/C catalyst from the kinetic to diffusion-limiting regions. We note that no rotation was applied to our bulky electrodes, suggesting that the Levich equation would not take effect[43]. As a result, the measured diffusion-limiting HOR currents on different catalysts deviate from the theoretical value. We measured a half-wave potential of 39 mV for Ni$_{5.2}$WCu$_{2.2}$, versus 44 mV for Pt/C catalyst. The observed striking HOR activity on Ni$_{5.2}$WCu$_{2.2}$ agrees well with the electrochemical

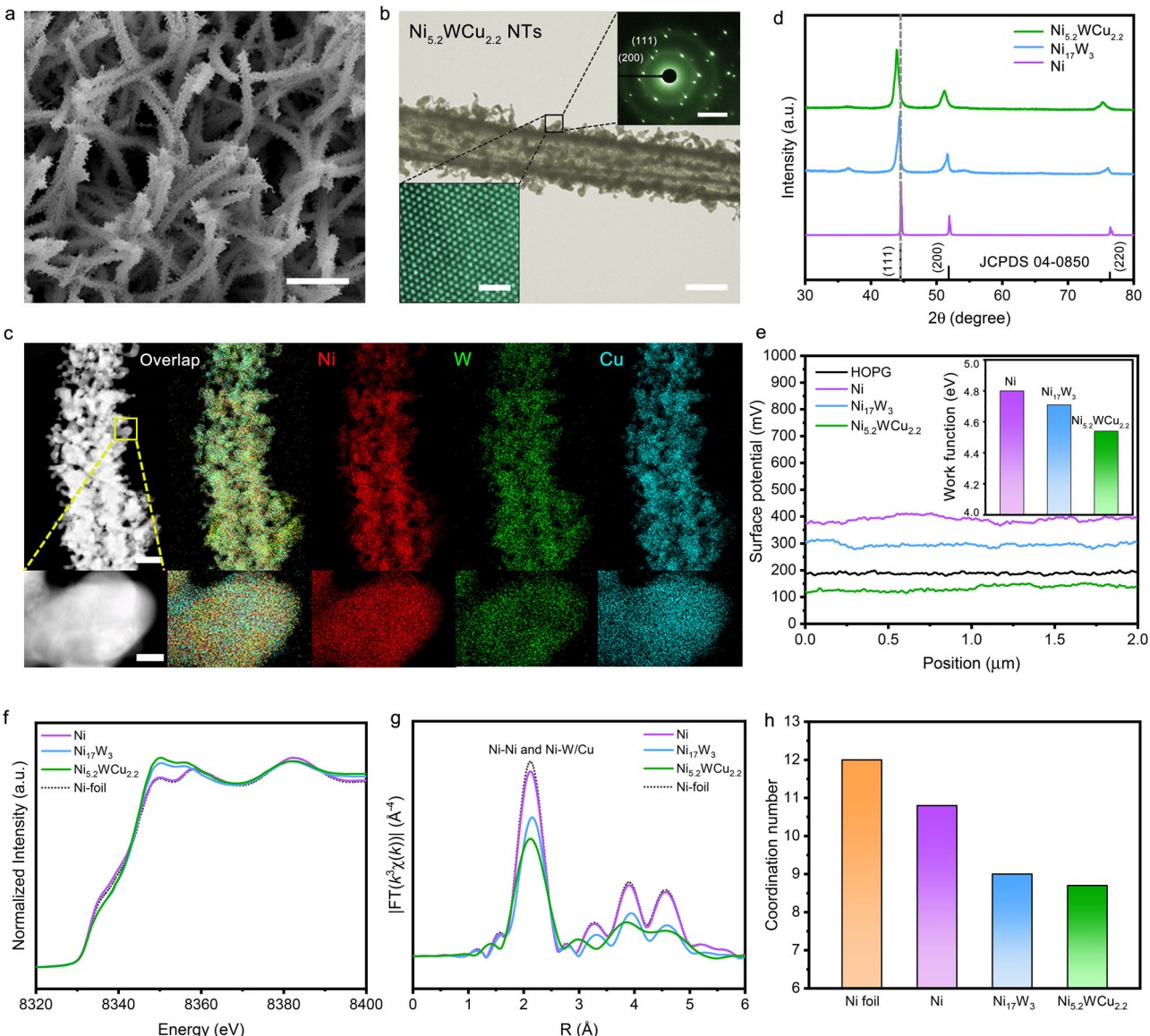

**Fig. 2 Characterization and structural analysis. a** SEM image of $Ni_{5.2}WCu_{2.2}$ alloy. Scale bar, 2 μm. **b** TEM image of $Ni_{5.2}WCu_{2.2}$ alloy. Scale bar, 250 nm. Insets show SAED pattern (up; scale bar, 5 1/nm) and the atomic-resolution HAADF-STEM image (down; scale bar, 1 nm), respectively. **c** STEM-EDX elemental mappings of $Ni_{5.2}WCu_{2.2}$ alloy, showing a uniform spatial distribution of Ni, W, and Cu. Scale bars, 200 nm (up) and 20 nm (down). **d** XRD patterns of Ni, $Ni_{17}W_3$ and $Ni_{5.2}WCu_{2.2}$, respectively. **e** Surface potential profiles along the white lines in Supplementary Figure 10 for Ni, $Ni_{17}W_3$, $Ni_{5.2}WCu_{2.2}$ and HOPG reference, respectively. Inset shows the resultant work function values for the studied catalysts. **f, g** Ni K-edge XANES spectra and corresponding Fourier transforms of $k^3$-weighted EXAFS spectra for Ni, $Ni_{17}W_3$, $Ni_{5.2}WCu_{2.2}$ and Ni foil reference. **h** The average coordination number in the first coordination shell of Ni atoms for Ni, $Ni_{17}W_3$, and $Ni_{5.2}WCu_{2.2}$ by EXAFS spectra curve fitting. The coordination number of Ni foil reference is 12.

impedance spectroscopy measurements that yield a small charge transfer resistance of ~3.0 Ohms at 30 mV overpotential (Supplementary Fig. 17).

In Fig. 3a, we show another important observation that we want to highlight in this work: that is, our nanostructured $Ni_{5.2}WCu_{2.2}$ catalyst can sustain HOR reactivity without deactivation up to 0.3 V versus RHE. Previously, most Ni-based HOR catalysts have been witnessed to lose activity rapidly above 0.1 V versus RHE owing to hydroxide formation[13,28]. Indeed, scanning LSV positively displays two pronounced peaks starting from 0.2 V versus RHE for $Ni_{17}W_3$ and 0.18 V versus RHE for Ni (Inset in Fig. 3a), which correspond to the formation of $Ni(OH)_2$ (ref. [44]). The limited tolerance to anode overpotentials poses significant risk of passivation under high current operation, leading to unsatisfied power output[2]. The stability window up to

0.3 V versus RHE for our $Ni_{5.2}WCu_{2.2}$ catalyst has not yet been achieved on previous PGM-free HOR catalysts (Fig. 3b), which would largely mitigate the risk of passivation under transient conditions.

We now extract the exchange current density ($j_0$)—the most inherent measure of HOR activity—of various catalysts from linear fitting of micro-polarization regions (−5 to 5 mV; Supplementary Fig. 18). The geometric $j_0$ of 11.36 mA $cm^{-2}$ for $Ni_{5.2}WCu_{2.2}$ largely surpasses the values of 1.23 mA $cm^{-2}$ for $Ni_{17}W_3$, 0.14 mA $cm^{-2}$ for freshly prepared Ni, and 4.99 mA $cm^{-2}$ for Pt/C catalyst (Fig. 3c, Supplementary Table 2). By fitting Bulter-Volmer equation in the Tafel regions, similar values were obtained (Fig. 3d, Supplementary Table 2 and Methods). To further quantify the intrinsic HOR activity, the $j_0$ is renormalized by the electrochemical active surface area (ECSA; see "Methods", Supplementary Fig. 19 and Table 3 for

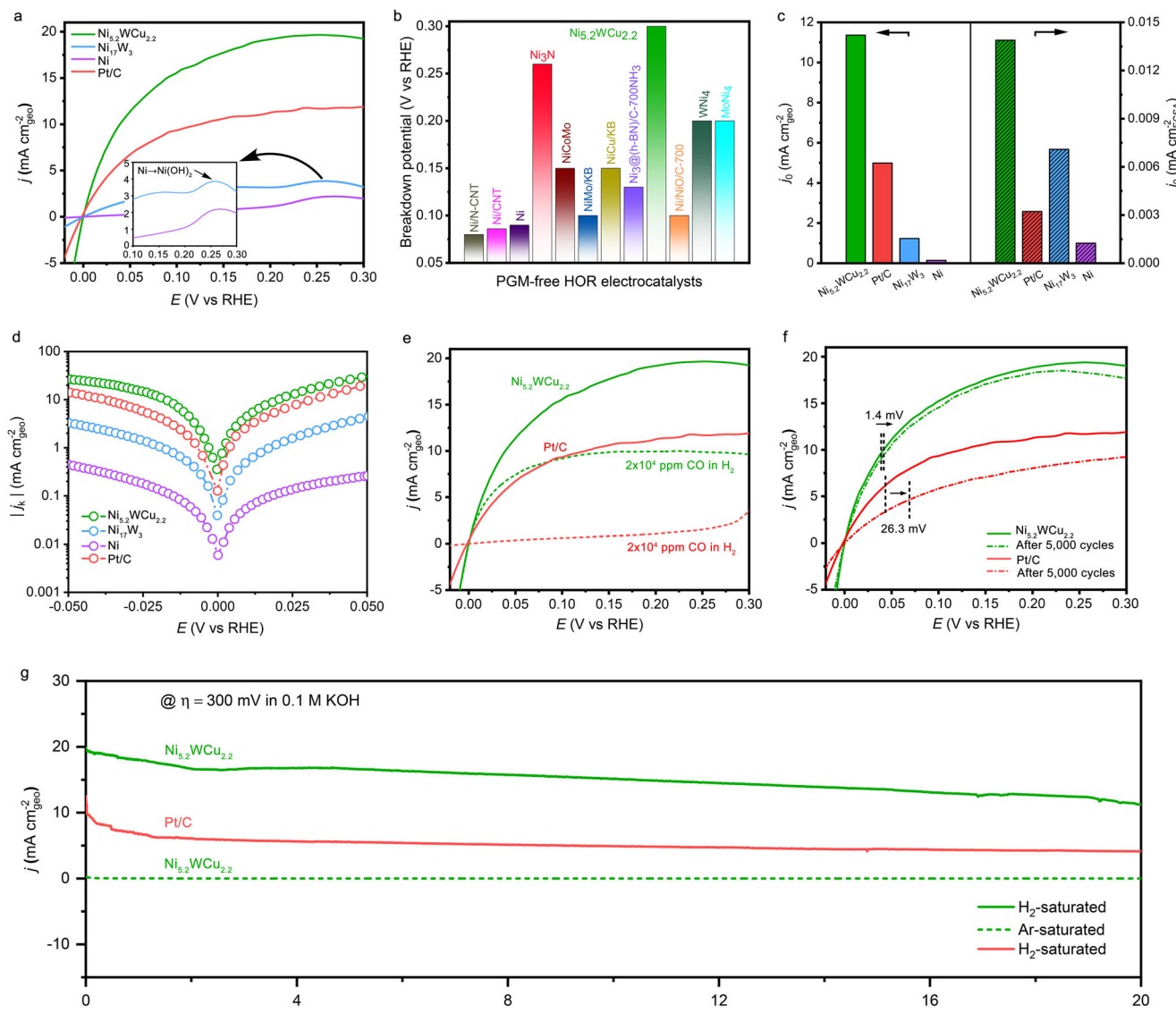

**Fig. 3 Electrocatalytic HOR performances. a** Polarization curves for the HOR on $Ni_{5.2}WCu_{2.2}$, $Ni_{17}W_3$, Ni, and commercial Pt/C catalyst measured in $H_2$-saturated 0.1 M KOH. Scan rate: $1\,mV\,s^{-1}$. Inset shows HOR curves at high anode potentials, showing the surface oxidation of $Ni_{17}W_3$ and Ni catalysts. **b** Comparison of breakdown potential of $Ni_{5.2}WCu_{2.2}$ with various PGM-free HOR catalysts reported previously. Breakdown potential means HOR stops at this potential. **c** Comparison of exchange current density ($j_0$) of various studied catalysts normalized by geometric areas (unpatterned) and ECSA (patterned), respectively. **d** HOR/HER Tafel plots of the kinetic current density on $Ni_{5.2}WCu_{2.2}$, $Ni_{17}W_3$, Ni, and Pt/C in $H_2$-saturated 0.1M KOH. **e** HOR polarization curves for $Ni_{5.2}WCu_{2.2}$ alloy and Pt/C catalyst in $H_2$-saturated 0.1M KOH with (dashed lines) and without (solid lines) the presence of 20,000 ppm CO. **f** HOR polarization curves for $Ni_{5.2}WCu_{2.2}$ alloy and Pt/C catalyst before (solid lines) and after (dashed lines) accelerated durability test, respectively. The durability test was performed at room temperature in $H_2$-saturated 0.1M KOH with the cyclic potential sweeping between −0.2 V to 0.2 V at a sweep rate of $200\,mV\,s^{-1}$. **g** Chronoamperometry ($j$ - t) responses recorded on $Ni_{5.2}WCu_{2.2}$ alloy and Pt/C catalyst at a 300 mV overpotential in $H_2$-saturated 0.1 M KOH at room temperature. Identical measurement on $Ni_{5.2}WCu_{2.2}$ alloy in Ar-saturated 0.1M KOH was also carried out for comparison.

details), yielding a value of $0.014\,mA\,cm^{-2}$ for $Ni_{5.2}WCu_{2.2}$, which is 1.96 times higher than $Ni_{17}W_3$ and 4.31 times higher than Pt/C catalyst (Fig. 3c). We also carried out the feed ratio-, temperature-, and time-dependent control experiments and unveiled that the optimized HOR activity was obtained on Ni-W-Cu alloy that was synthesized with Ni:W atomic ratio of 4:1 and then annealed at 500 °C for 1 h (Supplementary Figs. 20–24).

We next report that our ternary $Ni_{5.2}WCu_{2.2}$ alloy shows good resistance to CO poisoning. It is well known that PGM catalysts (such as Pt) are poisoned very rapidly in the presence of CO because of its preferential adsorption on Pt that blocks the active sites[45]. In Fig. 3e, we find that the Pt/C catalyst loses HOR activity completely with 20,000 ppm CO in the $H_2$ fuel. By stark contrast,

the $Ni_{5.2}WCu_{2.2}$ catalyst can retain high HOR activity with the same level of CO contamination, exhibiting its remarkable CO-tolerant ability. Besides, our results further show that the HOR activity decreases on both $Ni_{5.2}WCu_{2.2}$ and Pt/C catalysts when having 20,000 ppm $CO_2$ in the $H_2$ fuel (Supplementary Fig. 25). The detailed poison mechanism is unclear and needs further investigations.

Apart from activity and CO tolerance, another critical metric for the application of $Ni_{5.2}WCu_{2.2}$ alloy as a PGM-free anode is the long-term stability—especially the stability at high anode potentials. To assess this, we performed two sets of stability tests on our $Ni_{5.2}WCu_{2.2}$ alloy catalyst. We first performed accelerated durability tests (ADT) by applying linear potential sweeps between −0.2 and

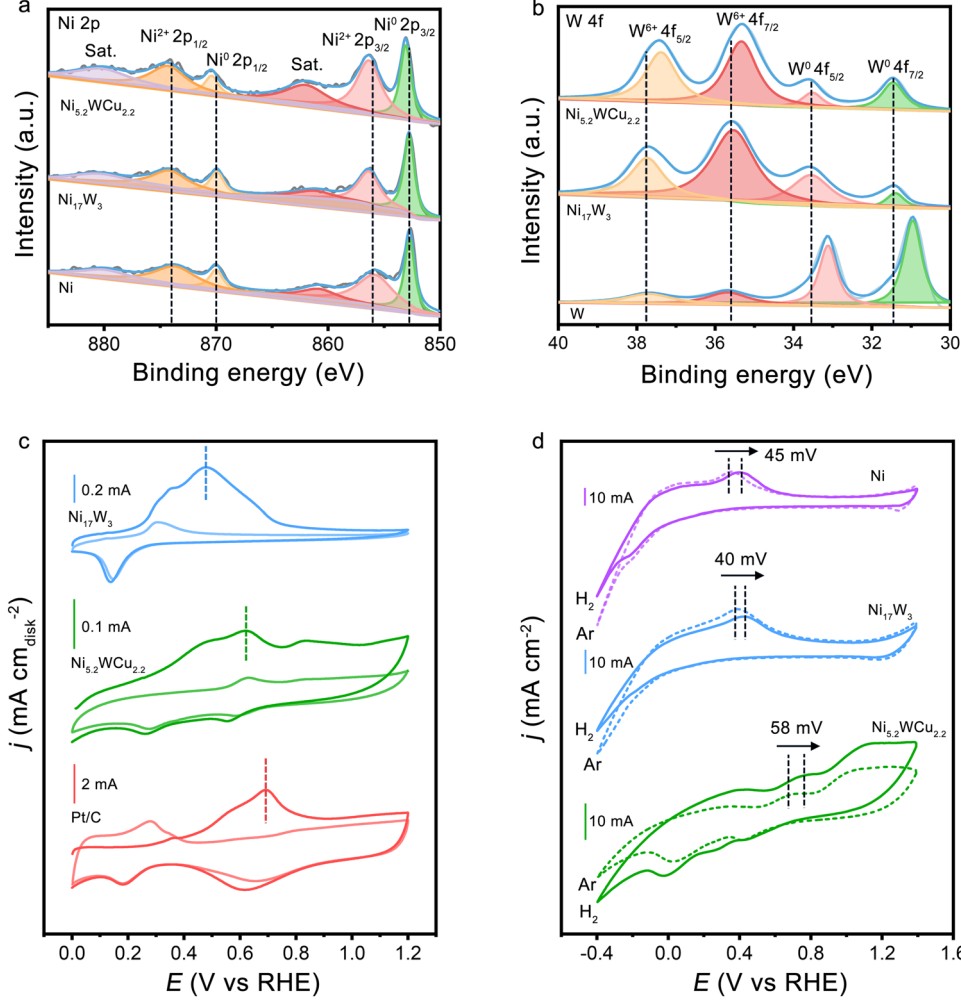

**Fig. 4 Surface analysis of different catalysts. a** Ni 2p XPS spectra of Ni$_{5.2}$WCu$_{2.2}$, Ni$_{17}$W$_3$, and Ni. **b** W 4f XPS spectra of Ni$_{5.2}$WCu$_{2.2}$, Ni$_{17}$W$_3$, and W. **c** CO-stripping measurements on Ni$_{5.2}$WCu$_{2.2}$, Ni$_{17}$W$_3$, and Pt/C catalyst in 0.1 M KOH electrolyte. Scan rate: 20 mV s$^{-1}$. Rotation speed: 1600 r.p.m. The light-colored curves in **c** show the second cycle of the measurements. **d** CV curves of Ni$_{5.2}$WCu$_{2.2}$, Ni$_{17}$W$_3$, and Ni in H$_2$- and Ar-saturated 0.1 M KOH, respectively. Scan rate: 50 mV s$^{-1}$.

0.2 V at 200 mV s$^{-1}$ in H$_2$-saturated 0.1 M KOH electrolyte. After 5,000 cycles, the Ni$_{5.2}$WCu$_{2.2}$ alloy shows a 1.4 mV increase in half-wave potential, versus 26.3 mV observed on Pt/C catalyst (Fig. 3f), implying that Ni$_{5.2}$WCu$_{2.2}$ alloy has stability better than Pt/C catalyst. Then, we evaluated the ability of Ni$_{5.2}$WCu$_{2.2}$ alloy to continuously catalyze the HOR using chronoamperometry ($j \sim t$) at a high anode potential of 0.3 V versus RHE. As shown in Fig. 3g, the current density ($\sim$20 mA cm$^{-2}$) generated from Ni$_{5.2}$WCu$_{2.2}$ decreases to 16.5 mA cm$^{-2}$ after 2 h, followed by slow degradation over the next 18 h. This degradation was likely caused by selective W leaching (Supplementary Fig. 26). By contrast, Pt/C catalyst delivers a current density of mere 10 mA cm$^{-2}$ at the beginning, which decreases progressively to about 4.3 mA cm$^{-2}$ over a 20 h of operation. After aggressive stability tests, the crystal phase, structure and composition of the Ni$_{5.2}$WCu$_{2.2}$ alloy remained almost unchanged (Supplementary Figs. 27–29), whereas Pt/C catalyst suffered from agglomeration and even detachment of Pt nanoparticles (Supplementary Fig. 30). These results reveal superior HOR stability of Ni$_{5.2}$WCu$_{2.2}$ alloy compared with Pt/C catalyst.

**Surface structure and chemistry of catalyst**. The formation of a ternary Ni$_{5.2}$WCu$_{2.2}$ alloy enables a HOR catalyst that shows high activity and stability at large anode potential, as well as good

resistance to CO poisoning. In this section, we study the structural and chemical characters that affect and determine these performances. X-ray photoelectron spectroscopy (XPS) analysis of the studied samples showed that Ni, Ni$_{17}$W$_3$, and Ni$_{5.2}$WCu$_{2.2}$ are all metallic in nature with certain surface oxidation (Fig. 4a and b). We note that some degree of surface oxidation is nearly inevitable when these Ni-based compounds are exposed to air, analogous to our previous observation[21]. XPS measurements also revealed that the Ni 2p binding energies of Ni$_{5.2}$WCu$_{2.2}$ shifted 0.3 eV to higher energy with respect to Ni$_{17}$W$_3$ (Fig. 4a). Meanwhile, the W 4f XPS of Ni$_{5.2}$WCu$_{2.2}$ showed a shift of the W core levels to lower binding energies compared with Ni$_{17}$W$_3$ (Fig. 4b). These results indicated electron donation from Ni to W once Cu participate in the alloy, leading to modulated electronic structure. Our Bader analyses further confirm such charge redistribution after incorporating Cu into the alloy structure (Supplementary Fig. 31).

To further probe the surface features, we conducted CO-stripping (CO electrooxidation) experiments on Ni$_{17}$W$_3$ and Ni$_{5.2}$WCu$_{2.2}$ with Pt/C catalyst as a reference. Results shown in Fig. 4c display that the oxidation of adsorbed CO (CO$^*$) occurs at 0.69 V for Pt/C catalyst, which is in agreement with prior reports[46]. By contrast, Ni$_{17}$W$_3$ oxidizes CO$^*$ at a much lower potential of 0.48 V, and Ni$_{5.2}$WCu$_{2.2}$ has a CO-stripping peak

located in the middle (0.62 V). These results indicate that both $Ni_{17}W_3$ and $Ni_{5.2}WCu_{2.2}$ have a much lower CO adsorption ability compared with the Pt/C catalyst, leading to the good CO-tolerance property observed in Fig. 3e. Moreover, in alkaline electrolytes, because the OH adsorption can facilitate the removal of the $CO^*$ (ref. [47]), our CO-stripping results thus offer additional surface information that $Ni_{17}W_3$ and $Ni_{5.2}WCu_{2.2}$ possess stronger OH adsorption than Pt/C catalysts following the order of $Ni_{17}W_3 > Ni_{5.2}WCu_{2.2} > Pt/C$.

Surface analysis of the studied samples was also carried out using cyclic voltammetry in 0.1 M KOH electrolyte that saturated by Ar and $H_2$, respectively. At a large sweep rate of 50 mV s$^{-1}$, the oxidation curve of $Ni_{5.2}WCu_{2.2}$ appears at higher anode potential than that of Ni and $Ni_{17}W_3$ (Fig. 4d), suggesting its superior oxidation-tolerant ability[27]. Intriguingly, we find that the oxidation peak of $Ni_{5.2}WCu_{2.2}$ in $H_2$-saturated 0.1 M KOH exhibits a 58 mV positive shift relative to that in Ar-saturated 0.1 M KOH (Fig. 4d). Similar positive shifts were also observed on Ni (45 mV) and $Ni_{17}W_3$ (40 mV) but with smaller values. The notable shift of oxidation peak in $H_2$- and Ar-saturated electrolytes likely result from the various degrees of $H_2$ adsorbed on these Ni-based catalysts, which indicates that the adsorption of $H_2$ decreases in the order of $Ni_{17}W_3 > Ni > Ni_{5.2}WCu_{2.2}$.

**Computational studies.** With these experimental information in hand, we now turn to perform density functional theory (DFT) to gain fundamental insight into mechanisms responsible for the excellent activity and stability, as well as the good CO-tolerance

property (see Methods). We constructed and optimized crystal models of $Ni_{5.2}WCu_{2.2}$(111), $Ni_{17}W_3$(111), Ni(111), and Pt(111) to represent the catalytic surfaces (see Methods; Supplementary Figs. 32–36). According to the DFT calculations, O-species prefer to adsorb on Ni (0.133 eV) and $Ni_{17}W_3$ (0.22 eV), whereas such adsorption on $Ni_{5.2}WCu_{2.2}$ (0.31 eV) is considerably weak (Fig. 5a). These results predict that the oxidation of $Ni_{5.2}WCu_{2.2}$ is more difficult to occur as compared to Ni and $Ni_{17}W_3$ (refs. [48,49]), consistent with our experimentally observed HOR stability at higher anode potential. Additionally, our DFT simulations reveal that Pt possesses a much higher CO adsorption strength (−1.78 eV) than that of $Ni_{17}W_3$ (−1.42 eV) and $Ni_{5.2}WCu_{2.2}$ (−1.55 eV) (Fig. 5b), which explain the marked CO-tolerance ability of $Ni_{5.2}WCu_{2.2}$ and agree well with above CO-stripping trend (Fig. 4c).

Although the HOR mechanism in alkaline electrolytes is still debated[6,50–52], recent research has gradually led to agreement that hydrogen binding energy (HBE) and OH binding energy (OHBE)/oxophilicity both serve as descriptor for HOR performance[27,50,53]. With DFT calculations (Fig. 5c), we revealed that although the HBE of Pt(111) is optimum, whereas the adsorption energy of OH is too weak on Pt(111). As to Ni(111) facet, the HBE value is −0.58 eV, which means a too strong H-binding that hampers HOR catalysis, agreeing with previous study[23]. Our DFT results further show that ternary $Ni_{5.2}WCu_{2.2}$ alloy yields a HBE close to Pt(111); moreover, the hydroxyl adsorption on $Ni_{5.2}WCu_{2.2}$(111) is significantly enhanced compared with Pt(111). Thus, this ternary-alloy-designed $Ni_{5.2}WCu_{2.2}$ gives rise to near-optimal HBE and OHBE that

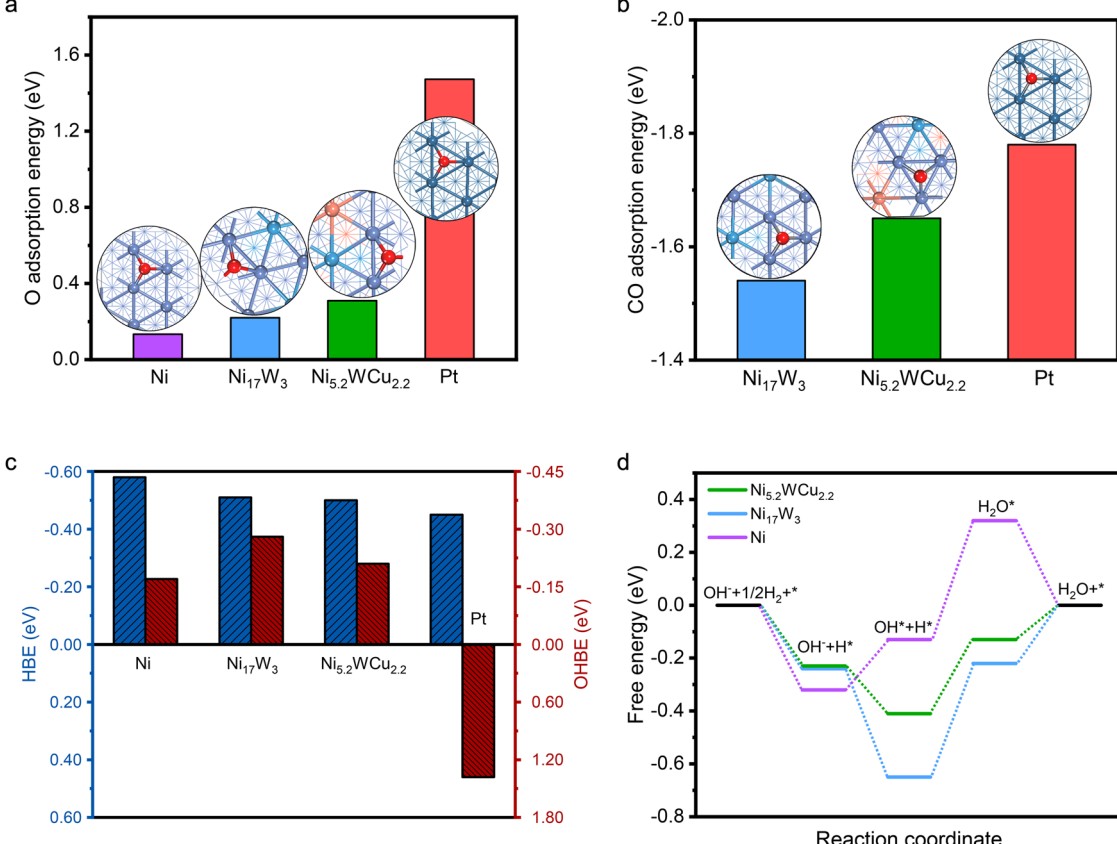

**Fig. 5 DFT calculation. a** O adsorption energy of $Ni_{5.2}WCu_{2.2}$, $Ni_{17}W_3$, Ni and Pt. The more positive value indicates the better anti-oxidation ability. **b** CO adsorption energy of $Ni_{5.2}WCu_{2.2}$, $Ni_{17}W_3$ and Pt. Insets in **a** and **b** presents corresponding catalyst models. Color labels: darkblue balls, Pt atoms; violet balls, Ni atoms; blue balls, W atoms; orange balls, Cu; black balls, C atoms; red balls, O atoms. **c** HBEs and OHBEs of $Ni_{5.2}WCu_{2.2}$, $Ni_{17}W_3$, Ni and Pt. **d** Free energy diagrams for reaction pathways on $Ni_{5.2}WCu_{2.2}$, $Ni_{17}W_3$ and Ni catalysts, respectively, revealing that the Volmer step is the rate-limiting step.

promote the HOR kinetics. We also computed the HBE and OHBE of $Ni_{17}W_3$ for comparison (Fig. 5c). Despite this catalyst exhibits stronger OHBE than Pt(111), its HBE is unfortunately strong, thus showing inferior HOR activity. The optimized HBE and OHBE that enabled by the synergy of Ni, W, and Cu in the ternary alloy can be further verified by our projected density of states (PDOS) analyses (Supplementary Fig. 37).

We note that our theoretically predicted trend of OH adsorption and CO adsorption perfectly match with CO-stripping results (Fig. 4c). Moreover, the free energy diagrams that we computed for the reaction pathways on various catalysts are presented in Fig. 5d. Results show that both H and OH adsorptions on $Ni_{17}W_3$ and $Ni_{5.2}WCu_{2.2}$ are exergonic, whereas the $H_2O$ formation and desorption steps on the two catalysts are endothermic. By contrast, H adsorption and $H_2O$ desorption on Ni are exergonic, but OH adsorption and $H_2O$ formation on Ni are endothermic. Our calculations thus predict that $H_2O$ formation (i.e., Volmer step) is the rate-determining step, showing energy barriers of 0.45, 0.43, and 0.28 eV for the Ni, $Ni_{17}W_3$ and $Ni_{5.2}WCu_{2.2}$ catalysts, respectively. The much lower energy barrier obtained on $Ni_{5.2}WCu_{2.2}$ contributes to its exceptional HOR performance. Altogether, our experimental and computational studies reveal that the synergy among Ni, W, and Cu—a multiple-element alloying effect—enables the optimized HBE and OHBE that improve the energetics for HOR, as well as the wide stability window up to 0.3 V versus RHE.

## Discussion

To conclude, this work demonstrates a multiple-element Ni-based alloy strategy for creating HOR catalyst that makes use of earth-abundant elements, showing exceptional activity and stability in alkaline electrolyte. This notable HOR performance can be explained by the alloying effect among nickel, tungsten and copper, which work in synergy to enable optimized hydrogen and hydroxyl bindings, as well as the improved resistance against surface oxidation. Our study calls for further exploration of multiple-element alloys composed of other cheap metals, thereby aiding the development of more efficient HOR catalysts for AEMFC anodes.

## Methods

**Synthesis of Cu(OH)₂ nanowires on Cu foam**. The $Cu(OH)_2$ NWs were grown on commercial Cu foam via a simple anodization method. Prior to the growth of nanowires, a piece of Cu foam (1 cm × 3 cm) was ultrasonically cleaned by hydrochloric acid solution, ethanol and deionized water, respectively, which was then immersed into a two-electrode system using Pt foil as the counter electrode in 1 M NaOH solution. The anodization was carried out at a constant current of 60 mA for 20 min to obtain $Cu(OH)_2$ nanowires on Cu foam.

**Synthesis of Ni₅.₂WCu₂.₂, Ni₁₇W₃ and Ni**. Firstly, $Ni(NO_3)_2 \cdot 6H_2O$ (581.6 mg), $(NH_4)_6H_2W_{12}O_{40} \cdot xH_2O$ (123.2 mg) and $CO(NH_2)_2$ (360.4 mg) were dissolved in 30 mL deionized water, and then transferred into a 50 mL Teflon-lined stainless-steel autoclave. The as-synthesized indigo $Cu(OH)_2$ NWs on Cu foam was immersed into the solution and placed against the wall of autoclave. Subsequently, the autoclave was maintained at 130 °C for 8 h. The foam was taken out and washed by deionized water several times to obtain $NiW-Cu(OH)_2$ NWs precursor. Finally, the precursor was annealed at 500 °C for 1 h with a heating rate of 5 °C under 5 % $H_2$/Ar atmosphere to obtain $Ni_{5.2}WCu_{2.2}$ alloys. The mass loading of $Ni_{5.2}WCu_{2.2}$ alloys was ~9.2 mg cm$^{-2}$ as determined by weighing the mass of Cu foam before anodization and after the annealing process.

By comparison, $Ni_{17}W_3$ and Ni precursors were directly grown on a clean Ni foam (1 cm × 3 cm) that was not anodized in prior. To obtain NiW precursors, Ni $(NO_3)_2 \cdot 6H_2O$ (581.6 mg), $(NH_4)_6H_2W_{12}O_{40} \cdot xH_2O$ (123.2 mg) and $CO(NH_2)_2$ (360.4 mg) were dissolved in 30 mL deionized water, and then transferred into a 50 mL Teflon-lined stainless-steel autoclave. The pure Ni foam was immersed into the solution and placed against the wall of autoclave. The Ni precursors were obtained under the same conditions without the addition of $(NH_4)_6H_2W_{12}O_{40} \cdot xH_2O$ (123.2 mg). The hydrothermal process was all performed under 130 °C for 8 h. Then the foam was taken out and washed by deionized water several times followed by an annealing process at 500 °C for 1 h with a heating rate of 5 °C under 5 % $H_2$/Ar atmosphere to obtain $Ni_{17}W_3$ and Ni nanoparticles.

**Material characterizations**. The obtained samples were examined by multiple analytic techniques. Optical microscope images were obtained with a polarizing microscope (Leica DM2700P, Germany) equipped with a Leica MC190 HD camera. The morphology of the samples was determined by SEM (Zersss Supra 40) and TEM (JEOL 2010F(s)). The STEM and HRTEM images, and EDX elemental mappings were taken on JEMARM 200 F Atomic Resolution Analytical Microscope with an acceleration voltage of 200 kV. The KPFM characterization was carried out with Atom Force Microscope (Dimension Icon). ICP-AES data was obtained by an Optima 7300 DV instrument. XRD was performed on a Japan Rigaku DMax-γA X-ray diffractometer with Cu Kα radiation ($λ = 1.54178$ Å). XPS was taken on an X-ray photoelectron spectrometer (ESCALab MKII) with an X-ray source (Mg Kα $hv = 1253.6$ eV).

**XAFS measurements**. The XAFS spectra (Ni K-edge) were collected at 1W1B station in Beijing Synchrotron Radiation Facility (BSRF). The storage rings of BSRF were operated at 2.5 GeV with an average current of 250 mA. Using Si(111) double-crystal monochromator, the data collection was carried out in transmission/fluorescence mode using ionization chamber. All spectra were collected in ambient conditions. The $k^3$-weighted EXAFS spectra were obtained by subtracting the post-edge background from the overall absorption and then normalizing with respect to the edge-jump step. Subsequently, $k^3$-weighted $χ(k)$ data of Ni K-edge were Fourier transformed to real (R) space using a hanning windows ($dk = 1.0$ Å$^{-1}$) to separate the EXAFS contributions from different coordination shells. To obtain the quantitative structural parameters around central atoms, least-squares curve parameter fitting was performed using the ARTEMIS module of IFEFFIT software packages[54].

**Electrochemical measurements**. A standard three-electrode set-up was applied to perform the HOR electrochemical measurements on the VSP-300 Potentiostat (Bio-Logic, France). Carbon rod was used as the counter electrode and all potentials were measured against an Ag/AgCl reference electrode (saturated in 3.0 M KCl) and converted to the RHE reference scale using following equation:

$$E(vs. RHE) = E(vs. Ag/AgCl) + 0.197 + (0.059 \times pH) \qquad (1)$$

The as-synthesized $Ni_{5.2}WCu_{2.2}$, $Ni_{17}W_3$ and Ni catalysts on Cu foam were directly used as working electrodes. After the annealing process, they were cut into 1 cm × 1 cm pieces for electrochemical tests. The commercial Pt/C was coated on a pure Cu foam (1 cm × 1 cm) with optimized loading of 1.5 mg cm$^{-2}$ (Supplementary Fig. 38). Before HOR measurements, the electrolyte (0.1 M KOH) was bubbled with $H_2$ gas for at least 30 min. The EIS measurement was performed at 30 mV overpotential and an amplitude of the sinusoidal voltage of 5 mV (frequency range: 100 kHz to 40 mHz). All the linear sweep voltammetry (LSV), cyclic voltammetry (CV), and chronoamperometry (CA) curves were iR-corrected.

The exchange current density ($j_0$) can be obtained by fitting kinetic current density ($j_k$) versus the overpotential ($η$) using the following Butler–Volmer equation:

$$j_k = j_0(e^{\frac{aF}{RT}η} - e^{\frac{-(1-a)F}{RT}η}) \qquad (2)$$

where $α$ is the charge transfer coefficient, $η$ is the overpotential, $R$ is the ideal gas constant (8.314 J mol$^{-1}$ K$^{-1}$), $T$ is the experimental temperature (298 K), and $F$ is the Faraday's constant (96,485 C mol$^{-1}$).

CO stripping was performed by holding the electrode potential at 0.1 V versus RHE for 10 min in the purged CO to adsorb CO on the metal surface, followed by Ar purging for another 30 min to remove residual CO in the electrolyte. The CO stripping current was obtained via cyclic voltammetry in a potential region from 0 to 1.2 V at a sweep rate of 20 mV s$^{-1}$.

**DFT calculations**. We carried out DFT calculations using the Vienna ab initio simulation package (VASP)[55,56] program with projector augmented wave (PAW)[57,58] method. The Perdew-Burke-Ernzerhof (PBE)[59] generalized gradient approximation (GGA) exchange-correlation functional was used throughout. A 500 eV plane-wave kinetic energy cutoff was chosen, and a 5 × 5 × 1 Monhorst-Pack k-point sampling was adopted for the structure relaxation. The convergence criterion for the electronic self-consistent iteration was set to be 10$^{-4}$ eV. A residual force threshold of 0.02 eV Å$^{-1}$ was set for geometry optimizations. The calculations were conducted on (111) surface of Ni, $Ni_{17}W_3$, $Ni_{5.2}WCu_{2.2}$, and Pt models. Hubbard $U$ corrections were applied to transition metal d-electrons and the values of $U$–$J$ parameters for Ni (3.80), W (6.20) and Cu (3.08) atoms were taken from the references[60–62]. The vacuum layer was set to be 15 Å to ensure the separation between slabs.

The key reaction steps in alkaline HOR:

$$H_2 + OH^- + * \rightarrow *H + H_2O + e^- \qquad (3)$$

$$*H + OH^- \rightarrow *H + *OH + e^- \qquad (4)$$

$$*H + *OH \rightarrow *H_2O \qquad (5)$$

$$*H_2O \rightarrow * + H_2O \qquad (6)$$

The Gibbs free energy changes are calculated as follows:

$$\Delta G_1 = G(*H) + G(H_2O) - G(*) - G(OH^-) - G(H_2) \qquad (7)$$

$$\triangle G_2 = G(*OH - *H) - G(*H) - G(OH^-) \qquad (8)$$

$$\triangle G_3 = G(*H_2O) - G(*OH - *H) \qquad (9)$$

$$\triangle G_4 = G(*) + G(H_2O) - G(*H_2O) \qquad (10)$$

The G values are calculated by:

$$G = H - T\Delta S = E_{DFT} + E_{ZPE} - TS \qquad (11)$$

$E_{DFT}$ is the total energy from the DFT calculation. $E_{ZPE}$ is the zero-point energy, $S$ is the entropy and $T$ is the temperature (298 K).

The adsorption energies ($\Delta E_{ad}$) for O and CO were calculated by the following equations:

The O adsorption energies were calculated by:

$$\Delta E_{O-ad} = E_{O@cat.} - E_{cat} - E_O \qquad (12)$$

The CO adsorption energies were calculated by:

$$\Delta E_{CO-ad} = E_{CO@cat.} - E_{cat} - E_{CO} \qquad (13)$$

The HBE and OHBE can be calculated by following equations:
The HBE were calculated by:

$$HBE = E_{H@cat.} - E_{cat} - E_H \qquad (14)$$

The OHBE were calculated by:

$$OHBE = E_{OH@cat.} - E_{cat} - E_{OH} \qquad (15)$$

Note: $E_{O@cat.}$, $E_{CO@cat.}$, $E_{H@cat.}$, and $E_{OH@cat.}$ represent the energies of metals or alloy slabs with the adsorbed O, CO, H, and OH species; the $E_{cat}$, $E_O$, $E_{CO}$, $E_H$, and $E_{OH}$ represent the energies of the metals or alloys slabs, the O atoms, the CO species, the H atoms, and OH species, respectively. Since the ground state of $O_2$ molecule is poorly described by DFT calculations, we thus used gas-phase $H_2O$ and $H_2$ as references to calculate the $E_O$ ($E_O = E_{H2O} - E_{H2}$). As to the $E_{OH}$, it can be obtained by $E_{OH} = E_{H2O} - 1/2\ E_{H2}$. We calculated the Gibbs free energy changes of $G(OH^-)$, $G(H_2)$ and $G(H_2O)$ according to a recent literature[63].

## Data availability

All experimental data within the article and its Supplementary Information are available from the corresponding author upon reasonable request.

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

## Acknowledgements

This work is supported by the National Basic Research Program of China (Grant 2018YFA0702001), the National Natural Science Foundation of China (Grants 21975237 and 51702312), the Anhui Provincial Research and Development Program (Grant 202004a05020073), the Fundamental Research Funds for the Central Universities (Grant WK2340000101), the USTC Research Funds of the Double First-Class Initiative (Grant YD2340002007), and the Recruitment Program of Global Youth Experts. Y.D. acknowledges the China Postdoctoral Science Foundation (2020M682008, 2020TQ0309).

## Author contributions

M.R.G. conceived and supervised the project. S.Q. and Y.D. performed the experiments, collected and analyzed the data. X.L.Z carried out the DFT calculations. X.S.Z. and J.F.Z. performed XPS measurements. L.-R.Z. collected and analyzed the XANES data. F.Y.G., P. P.Y., Z.Z.N., R.L., and Y.Y. helped with electrochemical data collection and analysis. M.R. G. and S.Q. co-wrote the manuscript. All authors discussed the results and commented on the manuscript.

## Competing interests

The authors declare no competing interests.
