## [Peer Review File · Nature Communications]

REVIEWER COMMENTS

Reviewer #1 (Remarks to the Author):

In this manuscript, Qin et al. reported Ni-W-Cu alloy HOR catalyst. The oxidation tolerance and stability of the catalyst are impressive. In my view, this manuscript is well written and suitable for publication in Nature Communications after the authors could consider some issues below.

1. In this work very high loading of Ni-W-Cu catalyst was employed (9.2 mg/cm²) while the Pt loading was only 0.3 mg/cm² (assuming that the loading values in Figure S31 referred to the total loading of Pt and carbon support). Thus, the intrinsic activity normalized to the mass of catalyst should also be listed for comparison. Meanwhile, with the high loading of this Ni-W-Cu catalyst, its mass activity and specific activity is lower than that of Ni-Mo alloy from the authors' recent published work (ref 20). Is there a possibility for further improvement of the measured intrinsic activity with optimized loading?
2. Since the measured surface area of Pt/C catalyst was mostly contributed by carbon support via BET method, j_0 of Pt/C catalyst normalized to this surface area is not meaningful as a benchmark. The specific activity of Pt/C is better to be exhibited as j_0 normalized to its electrochemical active surface area from hydrogen adsorption or CO stripping method. Additionally, if the electrochemical active surface area of Ni-W-Cu and Ni-W could be measured by CO stripping method, it is a better descriptor for density of active sites than surface area from BET method.
3. How to interpret that Ni-W-Cu and Pt/C catalysts in Figure 3a have different diffusion-limiting currents?
4. The definition of "breakdown potential" shown in Figure 3b should be clarified.

Reviewer #2 (Remarks to the Author):

The manuscript entitled 'Ternary nickel-tungsten-copper alloy rivals platinum for catalyzing alkaline hydrogen oxidation up to 0.3 volt versus reversible hydrogen electrode' by Qin et al. reported a novel electrocatalyst for HOR with the high performance and stability in alkali. The characterization results reveal that the synthesized electrocatalysts possess the hollow nanofiber morphology, which can benefit the catalytic process. The composition of the alloy has further been identified as Ni_{5.2}WCu_{2.2} based on the XRD, ICP-AES and EDX data. Using this novel electrocatalyst, a high anode potential up to 0.3 V vs RHE has been achieved, which is very impressive. Moreover, the catalysts can have high CO-tolerance and stability. Finally, the DFT calculations have been conducted to explain the experimental observations. The performance of this novel ternary alloy based HOR electrocatalyst is outstanding, which can greatly benefit the advance of fuel cells in alkaline. The experimental and characterization methods are reliable. However, more computational details are required before this paper can be accepted.

- 1) The XRD data reveal the change of the lattice constant of Ni metal after the introduction of W and Cu. To this end, the lattice constants used to construct the atomic model need to be provided in the supplementary information.
- 2) The work function derived from the UPS data can be confirmed by the theoretical result. Such comparison can also be used to justify the atomic models used in the DFT calculations.
- 3) The XPS data suggest that the electron donation from Ni to W and Cu upon alloying. The charge analysis of the atoms can confirm it.
- 4) Some computational details about the adsorption energies are missing. How did the authors define the energy of the adsorbate here? What are the values of $G(\text{OH}^-)$, $G(\text{H}_2)$ and $G(\text{H}_2\text{O})$? How to calculate HBE and OHBE shown in Fig. 5c.
- 5) The authors mentioned that the adsorption of OH^* can be used as the key descriptor to evaluate

the HOR performance of electrocatalysts. Why are the key reaction steps in alkaline HOR not related to the adsorption of OH?

6) From Supplementary Figures 26-30, it seems that a (4×4) surface cell was used to model Pt(111) and Ni(111) surface, respectively. However, a (3×3) surface cell was used to model the Ni₁₇W₃(111) and Ni_{5.2}W₂Cu_{2.2}(111), respectively. The coverage of the adsorbate may affect its adsorption energy. As such, the same coverage is required for the comparative study.

7) The introduction of the W and Cu reduces the symmetry of the surface. The authors need to study difference adsorption sites to find the most stable one. The authors may only provide the most stable adsorption configuration here. If it is true, the authors can provide all the adsorption properties in the Supplementary Information, which can justify the atomic models used here. For example, why does the oxygen atom interact with three surface Ni atoms on Ni₁₇W₃(111) surface while the oxygen atom interacts with two Ni and one W atoms on the Ni_{5.2}W₂Cu_{2.2}(111) surface shown in Supplementary Fig. 27?

8) The authors should explain how the synergy among Ni, W and Cu can optimize the HBE and OHBE. To address this issue, the electronic properties of the surface atoms, e.g. their PDOS and charge, need to be analyzed.

9) The format of some references is not correct. See Ln. 39, 228, 247.

10) There are some grammatical errors.

Reviewer #3 (Remarks to the Author):

Excellent work. The authors present a novel nickel-tungsten-copper catalyst highly active towards HOR in alkaline. The authors claim to achieve a high anode potential up to 0.3 V with a good stability. This work is interesting for the research community. I'd be happy to see this work published after the authors address some minor points:

1. Literature refs look not well balanced. Readers should be aware of the ample and intensive activities in the alkaline HOR field. While discussing and reviewing the HOR in alkaline, the authors must cite and discuss recent and VERY relevant studies done on, for instance, Ni-based HOR catalysts for alkaline media done by many other groups (out of my memory, I can think on Atanassov et al., Savinova et al., Dekel et al., Abruna et al., Yan et al., and many others).

2. The authors refer to hydroxide exchange membrane fuel cells (HEMFCs). Although this term is used for another 1-2 groups in the world, the common term of >95% of the publications is anion-exchange membrane fuel cells (AEMFCs). The reason for that is that eventually while working with ambient air (containing CO₂), the membrane conducts not only hydroxides but also (bi)carbonate anions. At any rate, again, although correct calling it HEMFCs, the most general and correct term may be AEMFCs. Also, the visibility of this term will be wider, which may be of interest for the authors.

3. Authors claim that "... there is no PGM-free HOR catalyst with operating stability 67 up to 0.3 V versus RHE has been reported thus far" – I have found at least one study (ACS Catalysis 9, 6837-6845, 2019; ref 27 in the manuscript) that shows Ni-Fe catalysts with negligible dissolution up to 0.7 V. I had the impression that may be many other similar studies. This is connected to my previous comment re. not a satisfactory literature review. Authors should expand their review of the literature to give a better picture of the field to the readers.

4. "we speculate that Cu may play an important role in HOR catalysis. Hence, we decide to introduce Cu to form ternary Ni-W-Cu alloy and explore its alkaline HOR property." – Although fear enough, the readers may want to get a more scientific explanation of why the authors believe copper is the best option.

5. Excellent results! The authors compare the performance of Ni, as the base case, with NiW, and NiWCu. Although this is very informative, the stoichs used are completely different. I find it difficult to understand the comparison between Ni₅W₂Cu₂ and Ni₁₇W₃. Somehow a better comparison will be adding Ni₁₇W₃Cu_x and Ni₅W, for instance. Also, why these specific compositions were selected is not clear.

6. At which temperature the Chronoamperometry responses were carried out in fig 3?

7. Although HBE is critical, some other studies indicate that OHBE may be also used as indicator for HOR in alkaline (see for instance, *Catalysts*, 8(10), 454, 2018). Readers may be very interested in this discussion.
8. Although the results are very clear and informative, the science behind is not clear. Why this approach does improve the stability of the Ni catalyst to a potential 0.3 V?
9. It will be of interest to show the oxygen map in Figure 2c.
10. Why the limiting current for the HOR in Figure 3a,e,f is too high compared to the theoretical value? What is the rotation rate used for HOR? Please check articles for theoretical limiting (among many others, <https://doi.org/10.1002/adfm.202002087>)
11. Typically, Ni-based catalysts are poised to a greater extent with CO₂ than CO. Therefore, it will be interesting and more helpful for the alkaline environment, where CO₂ cannot be avoided (see for instance <https://doi.org/10.1021/acscatal.6b00487>)
12. In Figure 3(c, d), the author may want to normalize the j_o with the electrochemical surface to get the intrinsic activity to avoid the effect of loading and particle size effect.
13. Including data of AEMFC test with the NiWCu best catalyst will be of great value.

We thank all the reviewers for their valuable comments and questions that help us significantly improve the revised manuscript.

REVIEWER REPORTS:

Reviewer #1 (Remarks to the Author):

In this manuscript, Qin et al. reported Ni-W-Cu alloy HOR catalyst. The oxidation tolerance and stability of the catalyst are impressive. In my view, this manuscript is well written and suitable for publication in Nature Communications after the authors could consider some issues below.

Response: We are very grateful for the reviewer's highly positive assessment of our work.

1. In this work very high loading of Ni-W-Cu catalyst was employed (9.2 mg/cm²) while the Pt loading was only 0.3 mg/cm² (assuming that the loading values in Figure S31 referred to the total loading of Pt and carbon support). Thus, the intrinsic activity normalized to the mass of catalyst should also be listed for comparison. Meanwhile, with the high loading of this Ni-W-Cu catalyst, its mass activity and specific activity is lower than that of Ni-Mo alloy from the authors' recent published work (ref 20). Is there a possibility for further improvement of the measured intrinsic activity with optimized loading?

Response: We appreciate the reviewer for his/her thoughtful comments and questions. Well, we synthesized the Ni_{5.2}WCu_{2.2} alloy by using the Cu foam as substrate and precursor, thus only bulky alloy can be obtained (please see **Figure 1** in our manuscript about the synthesis). We determined the monolithic alloy with catalyst loading of ~9.2 mg/cm². Because the Ni_{5.2}WCu_{2.2} phase was grown on Cu foam precursor, we are thus unable to control the loading by tuning synthetic parameters such as temperature, time and chemical ratio. That is to say, the loading of Ni_{5.2}WCu_{2.2} catalyst is fixed to be ~9.2 mg/cm² by our synthetic method. Thus, it is currently difficult for us to improve the intrinsic activity by optimizing the catalyst loading.

For the purpose of comparison, we sprayed commercial Pt/C catalyst onto the Cu foam with different loadings. We found that the Pt/C loading of 1.5 mg/cm² (or 0.3 mg_{Pt}/cm²) shows the optimum HOR activity. Further increasing the loading to 2.0 mg/cm² (or 0.4 mg_{Pt}/cm²), however, leads to decreased activity, indicating that higher catalyst loading causes mass transfer issue. This result suggests that the activity of Ni_{5.2}WCu_{2.2} alloy might be further improved if the loading of 9.2 mg/cm² can be reduced owing to the modified mass transfer ability. We also provided the intrinsic activity that normalized to the mass of our catalysts are given in **Table R1** for comparison.

Table R1. The mass activities and ECSA-normalized j_0 of different catalysts.

Catalysts	Mass activity @ = 50 mV (mA mg ⁻¹ _{Ni(Pt)})	Mass- normalized j_0 (mA mg ⁻¹ _{Ni(Pt)})	ECSA- normalized j_0 (mA cm ⁻² _{cat.})
Ni _{5.2} WCu _{2.2}	2.55	2.54	0.014
Ni ₁₇ W ₃	0.93	0.64	0.0071
Ni	0.25	0.14	0.0013
Pt/C	22.4	16.63	0.0032

With above considerations in mind, we are now exploring methods to synthesize high-quality Ni_{5.2}WCu_{2.2} powder catalyst, which would allow us to optimize the loading for improved performance. Moreover, the Ni_{5.2}WCu_{2.2} powder catalyst could also permit the RDE studies, which will enable a fair comparison with Ni-Mo powder alloy developed by us previously (*ref.* 21 in the revised manuscript).

2. Since the measured surface area of Pt/C catalyst was mostly contributed by carbon support via BET method, j_0 of Pt/C catalyst normalized to this surface area is not meaningful as a benchmark. The specific activity of Pt/C is better to be exhibited as j_0 normalized to its electrochemical active surface area from hydrogen adsorption or CO stripping method. Additionally, if the electrochemical active surface area of Ni-W-Cu and Ni-W could be measured by CO stripping method, it is a better descriptor for density of active sites than surface area from BET method.

Response: We thank the reviewer for the insightful comments and suggestion. We know that the best way to evaluate the intrinsic electrochemical activity of a catalyst is to calculate its specific activity based on its electrochemically active surface area (ECSA). However, the ECSA values are difficult to obtain for many non-noble metals: they cannot be calculated using the classic hydrogen under-potential deposition (UPD) like commonly done for Pt because no obvious hydrogen adsorption occurs prior to H₂ evolution. Moreover, the CO stripping method for calculating the ECSA is commonly applicable to some noble metals (Pt, Pd, *etc.*). Therefore, as to the non-noble catalysts, we have to turn to other methods to evaluate the ECSAs.

Because our synthetic strategy gave rise to the large Ni_{5.2}WCu_{2.2} monolith (see **Figure 1** in the revised manuscript). We thus turn to find a method that could be used to determine the ECSA of such bulky catalytic materials. Previous studies reveal that the ECSA values of monolithic catalysts can be determined from the double-layer capacitance according to equation $ECSA = C_{dl}/C_s$ (where C_s is the specific capacitance of the catalyst or the capacitance of smooth planar surface per unit area) that suggested by McCrory, Peters and Jaramillo (*J. Am. Chem. Soc.* **2013**, 135, 16977). For examples,

Sun and co-workers used this method to calculate the ECSA of bulky Ni₃Ni/Ni/Ni-foam catalyst (*Nat. Commun.* **2018**, 9, 4531-4540) as well as Co₂N/Co/Cobalt-foam catalyst (*ACS Energy Lett.* **2019**, 4, 1594-1601).

Thus, we calculated the ECSA values of our studied catalysts from the double-layer capacitance according to equation $ECSA = C_{dl}/C_s$. For the estimate of surface area, we adopted the general specific capacitance of $C_s = 11 \mu\text{F cm}^{-2}$ based on typical reported value (*Angew. Chem. Int. Ed.* **2019**, 58, 10644-10649). As shown in **Figure R1** below, our measurements give ECSA values of 818 cm² for Ni_{5.2}WCu_{2.2} monolith, 173 cm² for Ni₁₇W₃, 64 cm² for Ni, and 1582 cm² for Pt/C. On the basis of the more accurate ECSA values, we thus could offer a fair comparison of our studied catalysts that shown in **Figure R1f**.

In the new version of our paper, we replaced the BET-normalized j_0 with the ECSA-normalized one (see **Figure 3c** in the revised manuscript) and offered some discussions over there.

Figure R1. ECSA measurements. a-d CV curves of $\text{Ni}_{5.2}\text{W}_3\text{Cu}_{2.2}$ (a), Ni_{17}W_3 (b), Ni (c), and Pt/C (d) collected at various scan rates ranging from 4 to 20 mV s^{-1} in CH_3CN with 0.15 M KPF_6 . e The linear fitting of scan rate versus j (the difference between the anodic and cathodic current densities at open circuit potential). f Comparison of exchange current density (j_0) of various studied catalysts normalized by geometric areas (unpatterned) and ECSA (patterned), respectively.

3. How to interpret that Ni-W-Cu and Pt/C catalysts in Figure 3a have different diffusion-limiting currents?

Response: We thank the reviewer for this very thoughtful and excellent question! Actually, we also questioned this phenomenon when obtaining these HOR curves at the first time, which, however, was highly repeatable and thus is definitely correct. Now, we elucidate the underlying reasons that lead to this phenomenon.

Generally, for the RDE test, the catalysts tend to have the same diffusion limiting currents at a fixed rotating speed according to the Levich equation (*Analytical Chemistry*, **1962**, 34, 164): $i_l = 0.62nFAD^{2/3}\nu^{-1/6}c_0\omega^{1/2} = ABC_0\omega^{1/2}$, where i_l is the limiting current, D is the diffusivity of hydrogen in electrolyte, n is the number of electrons transferred in HOR ($n = 2$), A is the area of the electrode, ν is the kinematic viscosity of the electrolyte and c_0 is the solubility of H_2 in electrolyte, ω is the rotation rate and $B = 0.62nFD^{2/3}\nu^{-1/6}$. The equation reveals that the value of i_l depends solely on the rotation speed ω . **It should be noted that the Levich equation can not be applied when the value of ω is small. This is because that a small ω would cause the hydrodynamic boundary layer becomes very large as compared to the disk radius, thus the assumption used in derivation of the Levich equation no longer holds** (*J. Electrochem. Soc.* **2015**, 162, F1470-F1481).

As discussed above, our synthetic strategy gave rise to the bulky $Ni_{5.2}WCu_{2.2}$ monolithic catalyst (see **Figure 1** in the revised manuscript). Thus, we directly used the $Ni_{5.2}WCu_{2.2}$ monolith and the Pt/C-modified Cu foam as electrodes for HOR study. **That is to say, no rotation can be applied to these bulky electrode (i.e., $\omega = 0$), leading to the inapplicability of the Levich equation here.** Additionally, Galus and co-workers have demonstrated that RDE without rotating will approach unshielded liner diffusion conditions, yielding a finite limiting current (*Analytical Chemistry*, **1962**, 34, 164). We thus conclude that the bulky HOR electrodes without noticeable rotation led to the different diffusion-limiting currents, where the Levich equation does not work. We note that the different diffusion-limiting currents were also observed on other bulky HOR electrodes reported previously, such as the $Ni_3N/Ni/Ni$ -foam and Pt/Ni-foam catalysts (*Nat. Commun.* **2018**, 9, 4531).

On the basis of the reviewer's question, we have added some discussions in our revised manuscript to clarify this aspect.

4. The definition of "breakdown potential" shown in Figure 3b should be clarified.

Response: Thanks for this comment. Often, the term of "breakdown potential" is used in corrosion chemistry, which means at such potential, active electrochemical reaction occurs, leading to the corrosion or repassivation of the material surface (see *npj Materials Degradation* **2019**, 3, 22). Here, this term was used to express the potential at which the HOR reaction starts to terminate but another oxidation reaction happens (e.g., the self-oxidation of PMG-free HOR catalyst itself). We note that this term was also used previously by Hu and co-workers for showing the potential at which their

Ni₃N/C HOR catalyst loses HOR activity (*Angew. Chem. Int. Ed.* **2019**, 58, 7445-7449). Following your suggestion, we have added some words in the new version of our paper to clarify this terminology.

Reviewer #2 (Remarks to the Author):

The manuscript entitled ‘Ternary nickel-tungsten-copper alloy rivals platinum for catalyzing alkaline hydrogen oxidation up to 0.3 volt versus reversible hydrogen electrode’ by Qin et al. reported a novel electrocatalyst for HOR with the high performance and stability in alkali. The characterization results reveal that the synthesized electrocatalysts possess the hollow nanofiber morphology, which can benefit the catalytic process. The composite of the alloy has further been identified as Ni_{5.2}WCu_{2.2} based on the XRD, ICP-AES and EDX data. Using this novel electrocatalyst, a high anode potential up to 0.3 V vs RHE has been achieved, which is very impressive. Moreover, the catalysts can have high CO-tolerance and stability. Finally, the DFT calculations have been conducted to explain the experimental observations. The performance of this novel ternary alloy based HOR electrocatalyst is outstanding, which can greatly benefit the advance of fuel cells in alkaline. The experimental and characterization methods are reliable. However, more computational details are required before this paper can be accepted.

Response: We greatly appreciate the reviewer’s high praise on the new ternary Ni_{5.2}WCu_{2.2} alloy HOR catalyst reported in this work.

1) The XRD data reveal the change of the lattice constant of Ni metal after the introduction of W and Cu. To this end, the lattice constants used to construct the atomic model need to be provided in the supplementary information.

Response: We thank the reviewer for nice suggestion on this aspect. The lattice constants used to construct the Ni, Ni₁₇W₃, and Ni_{5.2}WCu_{2.2} are “a=b=c=3.5174Å, α=β=γ= 90°”, “a=b=c=3.6464Å, α=β=γ= 90°”, and “a=b=c=3.7201Å, α=β=γ= 90°”, respectively, which have been added in the revised Supplementary Information.

2) The work function derived from the UPS data can be confirmed by the theoretical result. Such comparison can also be used to justify the atomic models used in the DFT calculations.

Response: We thank the reviewer for the valuable comments and suggestion. In our original manuscript, we measured the work function of our studied catalysts by UPS. On the basis of your suggestion, we calculated the work functions of Ni(111), Ni₁₇W₃(111) and Ni_{5.2}WCu_{2.2}(111) according to the following equation:

$$\phi = E_{\text{vac}} - E_{\text{F}}$$

Where E_{vac} is the electrostatic potential of the vacuum level, and E_{F} is the Fermi energy. As shown in **Figure R2**. The calculated work functions for Ni(111), $\text{Ni}_{17}\text{W}_3(111)$ and $\text{Ni}_{5.2}\text{WCu}_{2.2}(111)$ surfaces were 4.8042 eV, 4.7683eV, and 4.509 eV, respectively.

We note that the calculated work functions are quite different from that measured by UPS. **We omitted previously that the proper samples used for UPS measurements should be high-quality thin films (*Appl. Phys. Lett.* 2009, 95, 183303).** To measure the work functions more precise, we thus turn to use Kelvin probe force microscopy (KPFM), which are widely applied to probe the local work functions of materials (*J. Am. Chem. Soc.* 2014, 136, 8875-8878; *Nano Lett.* 2011, 11, 3755-3758). We used the clean highly oriented pyrolytic graphite (HOPG) with work function value of 4.6 eV as a reference (*Rev. Sci. Instrum.* 2018, 89, 043702). **Figure R3 and Figure R4a** show the measured contact potential difference (CPD) and surface potential values of different studied catalysts. Based on these results, we calculated the work function values of 4.54 for $\text{Ni}_{5.2}\text{WCu}_{2.2}$, 4.71 eV for Ni_{17}W_3 and 4.80 eV for Ni (**Figure R4b**), respectively. **These experimental work function values perfectly agree with our DFT predicted values of 4.509 eV for $\text{Ni}_{5.2}\text{WCu}_{2.2}$, 4.7683eV for Ni_{17}W_3 and 4.8042 eV for Ni, respectively.**

In the new version of our paper, we updated the work function of the different catalysts with our KPFM results and provided some discussion over there.

Figure R2. Electrostatic potentials of Ni(111) surface (a), $\text{Ni}_{17}\text{W}_3(111)$ surface (b), and $\text{Ni}_{5.2}\text{WCu}_{2.2}(111)$ surface (c).

Figure R3. The CPD images of $\text{Ni}_{5.2}\text{W}\text{Cu}_{2.2}$ (a), Ni_{17}W_3 (b), Ni (c), and HOPG (d), respectively.

Figure R4. a, The surface potential profiles of Ni, Ni_{17}W_3 , $\text{Ni}_{5.2}\text{W}\text{Cu}_{2.2}$, and HOPG, respectively. **b,** The work functions derived from KPFM analysis for Ni, Ni_{17}W_3 , and $\text{Ni}_{5.2}\text{W}\text{Cu}_{2.2}$, respectively.

3) The XPS data suggest that the electron donation from Ni to W and Cu upon alloying. The charge analysis of the atoms can confirm it.

Response: We thank the reviewer for the good suggestion. Our XPS analyses on Ni 2p and W 4f reveal that W donates electrons to Ni in the Ni_{17}W_3 binary alloy (see **Figures 4a and b** in our revised manuscript). As to the $\text{Ni}_{5.2}\text{W}\text{Cu}_{2.2}$ ternary alloy, the XPS analyses show that both Ni and W

donate electrons to Cu (see **Figures 4a and b** in the revised manuscript). Following the reviewer's suggestion, we further carried out the Bader charge analysis to confirm it theoretically. As shown in **Figure R5** below, both Ni and W in the $\text{Ni}_{5.2}\text{W}\text{Cu}_{2.2}$ ternary alloy lose electrons, while the electrons on Cu increase, indicating a marked charge redistribution *via* the incorporation of Cu in the alloy. Therefore, our Bader charge results confirm the electrons donation from Ni and W to Cu, in consistent with the XPS analyses.

We have added the new computational results in the revised version of our paper and provided some discussions therein.

Figure R5. The Bader charge analysis. The positive values represent the atoms get electrons from other atoms, whereas the negative values indicate electron donation to other atoms.

4) Some computational details about the adsorption energies are missing. How did the authors define the energy of the adsorbate here? What are the values of $G(\text{OH})$, $G(\text{H}_2)$ and $G(\text{H}_2\text{O})$? How to calculate HBE and OHBE shown in Fig. 5c.

Response: We thank the reviewer for reminding us of these missing aspects regarding the DFT calculations. Here we clarify these aspects in detail as follows:

(1) The 'adsorption energy' can be defined as the decreasing energy while two materials are combined under the adsorption process in which an atom, ion, or molecule (adsorbate) is attached to the surface of a solid (adsorbent) (<https://www.materialsquare.com/blog/10-adsorption-energy-and-surface-energy-obtained-through-slab-structure>). The adsorption energies (ΔE_{ad}) for O and CO were calculated by the following equations:

The O adsorption energies were calculated by:

$$\Delta E_{O\text{-ad}} = E_{O@cat.} - E_{cat} - E_O$$

The CO adsorption energies were calculated by:

$$\Delta E_{CO\text{-ad}} = E_{CO@cat.} - E_{cat} - E_{CO}$$

(2) Also, the HBE and OHBE can be calculated by following equations:

The HBE were calculated by:

$$HBE = E_{H@cat.} - E_{cat} - E_H$$

The OHBE were calculated by:

$$OHBE = E_{OH@cat.} - E_{cat} - E_{OH}$$

Note: the $E_{O@cat.}$, $E_{CO@cat.}$, $E_{H@cat.}$, and $E_{OH@cat.}$ represent the energies of metals or alloy slabs with the adsorbed O, CO, H, and OH species; the $E_{cat.}$, E_O , E_{CO} , E_H , and E_{OH} represent the energies of the metals or alloys slabs, the O atoms, the CO species, the H atoms, and OH species, respectively. Since the ground state of O_2 molecule is poorly described by DFT calculations, we thus used gas-phase H_2O and H_2 as references to calculate the E_O ($E_O = E_{H_2O} - E_{H_2}$). As to the E_{OH} , it can be obtained by $E_{OH} = E_{H_2O} - 1/2 E_{H_2}$.

(3) We calculated the Gibbs free energy changes of $G(OH^-)$, $G(H_2)$ and $G(H_2O)$ according to a recent literature (*Nature* **2016**, 537, 382-386; see **Table R2** below). At the standard temperature (298.15 K) and pressure (1 atm), the values of $G(H_2)$ and $G(H_2O)$ are -6.937 eV and -14.32 eV, respectively. Thus, the $G(OH^-) = G(H_2O) - 1/2 G(H_2) = -10.85$ eV.

Table R2. Free energy corrections for gas-phase species (eV).

Species	E	ZPE	T S	G
H₂O	-14.214	0.564	0.67	-14.32
H₂	-6.771	0.268	0.434	-6.937
CO	-14.775	0.132	0.668	-15.311

In view of the reviewer's questions, we have clarified these aspects very carefully in our revised manuscript, which we believe are much better to read.

5) The authors mentioned that the adsorption of OH^* can be used as the key descriptor to evaluate the HOR performance of electrocatalysts. Why are the key reaction steps in alkaline HOR not related to the adsorption of OH ?

Response: We thank the reviewer's helpful question that points out the possible source of confusion. Indeed, the adsorption of OH in our original manuscript was not included in the detailed reaction pathways. To be more precise, we thus re-calculated the HOR reaction pathways on the Ni, Ni₁₇W₃ and Ni_{5.2}WCu_{2.2} catalysts (*Nat. Energy* **2020**, 5, 891-899, *ACS Catal.* **2020**, 11751-11757, *Nano Lett.* **2020**, 20, 3442-3448). As shown in **Figure R6** below, both the H and OH adsorptions on Ni₁₇W₃ and Ni_{5.2}WCu_{2.2} are exergonic, whereas the H₂O formation and desorption steps on Ni₁₇W₃ and Ni_{5.2}WCu_{2.2} are endothermic. By contrast, the H adsorption and H₂O desorption on Ni are exergonic, whereas the OH adsorption and H₂O formation on Ni are endothermic. On the basis of these results, we concluded that the H₂O formation step (*i.e.*, Volmer step) is the rate-determining step for the Ni, Ni₁₇W₃ and Ni_{5.2}WCu_{2.2} catalysts. Our calculations reveal that the energy barriers of the Volmer step are 0.45, 0.43, and 0.28 eV for the Ni, Ni₁₇W₃ and Ni_{5.2}WCu_{2.2} catalysts, respectively. Hence, the much lower energy barrier of mere 0.28 eV for the rate-determining step on the Ni_{5.2}WCu_{2.2} catalyst resulted in its remarkable HOR performance.

In the new version of our paper, we updated the **Figure 5** with these new results and offered some discussions therein.

Figure R6. Free energy diagrams for the reaction pathways on Ni_{5.2}WCu_{2.2}, Ni₁₇W₃ and Ni catalysts, respectively.

6) From Supplementary Figures 26-30, it seems that a (4×4) surface cell was used to model Pt(111) and Ni(111) surface, respectively. However, a (3×3) surface cell was used to model the Ni₁₇W₃(111) and Ni_{5.2}WCu_{2.2}(111), respectively. The coverage of the adsorbate may affect its adsorption energy. As such, the same coverage is required for the comparative study.

Response: We thank the reviewer for pointing this out to us. We are sorry for omitting this aspect when constructing the models. In the new version of our paper, we created a (3×3) surface cell to

model the Ni(111) and Pt(111) surfaces (see **Figures R8 and R9** below). Accordingly, the adsorption energies of the O, OH, H, and CO species were also re-calculated, which only show slight difference with that on (4×4) surface cell. We have updated the new models and the related computational results in the revised Supplementary Information.

Figure R7. a-d Models of Ni(111) and O, H, and OH adsorbed on Ni(111) surface, respectively.

Figure R8. a-e Models of Pt(111) and O, H, OH, and CO adsorbed on Pt(111) surface, respectively

7) The introduction of the W and Cu reduces the symmetry of the surface. The authors need to study difference adsorption sites to find the most stable one. The authors may only provide the most stable adsorption configuration here. If it is true, the authors can provide all the adsorption properties in the Supplementary Information, which can justify the atomic models used here. For example, why does the oxygen atom interact with three surface Ni atoms on $Ni_{17}W_3(111)$ surface while the oxygen atom interacts with two Ni and one W atoms on the $Ni_{5.2}W_{Cu_{2.2}}(111)$ surface shown in Supplementary Fig. 27?

Response: Indeed, we had carried out the calculations once the most stable adsorption configuration was achieved, and we fully agree with the reviewer that all potential adsorption properties are necessary to better justify the atomic models. During the revision on the computational part, we considered that W sites might not be the suitable adsorption sites because selective surface W leaching happens during the electrochemical tests (see **Supplementary Figure 26** in the revised Supplementary information). Instead, we believe that W acts as a synergy in the alloy catalyst.

With above considerations in mind, we carefully studied all possible configurations that enable the adsorption of H, OH, CO and O for $Ni_{5.2}W_{Cu_{2.2}}$ and $Ni_{17}W_3$ catalysts (**Figures R10, R11, R12, and R13**), respectively. By using the most stable adsorption configuration, we have modified the computations and the results were updated in the **Figure 5** in the revised manuscript, which follow the same trend with our previous conclusion.

Figure R9. Models of all possible adsorption configurations on $\text{Ni}_{5.2}\text{W}_3\text{Cu}_{2.2}$ and Ni_{17}W_3 for HBE. The red label represents the most stable configuration.

Figure R10. Models of all possible adsorption configurations on $\text{Ni}_{5.2}\text{WCu}_{2.2}$ and Ni_{17}W_3 for OHBE. The red label represents the most stable configuration.

Figure R11. Models of all possible adsorption configurations on $\text{Ni}_{5.2}\text{WCu}_{2.2}$ and Ni_{17}W_3 for CO adsorption energies. The red label represents the most stable configuration.

Figure R12. Models of all possible adsorption configurations on Ni_{5.2}WCu_{2.2} and Ni₁₇W₃ for O adsorption energies. The red label represents the most stable configuration.

8) The authors should explain how the synergy among Ni, W and Cu can optimize the HBE and OHBE. To address this issue, the electronic properties of the surface atoms, e.g. their PDOS and charge, need to be analyzed.

Response: We thank the reviewer for the suggestion on improving the interpretation about how the synergy among Ni, W and Cu optimizes the HBE and OHBE. We agree on the reviewer that PDOS could be a suitable way to address this concern. Following your suggestion, we calculated the PDOS of nickel atoms for Ni, Ni₁₇W₃ and Ni_{5.2}WCu_{2.2}, respectively. As shown in **Figure R13a** below, the *d*-band center of Ni atom in single Ni shifts a higher electron density (-2.79 eV) away from the Fermi level, whereas the *d*-band center of Ni atoms relative to the Fermi level are -2.31 eV for Ni₁₇W₃ and -2.51 eV for Ni_{5.2}WCu_{2.2} catalyst. These results thus indicate that the OH adsorption on catalysts decreases in the order of Ni₁₇W₃ > Ni_{5.2}WCu_{2.2} > Ni (*Angew. Chem. Int. Ed.* **2018**, 57, 16421-16425; *Sci. Adv.* **2020**, 6, eaaw8113; *Angew. Chem. Int. Ed.* **2019**, 58, 7445-7449), suggesting that the synergy among Ni, W and Cu gives rise to a more suitable OH adsorption on Ni_{5.2}WCu_{2.2}.

Additionally, our calculations (**Figure R13b**) reveal the higher PDOS of single Ni at the Fermi level, which indicates its better charge transfer kinetics and thus a stronger H binding (*Nat. Mater.* **2019**, 18, 1309-1314). By contrast, the Ni_{5.2}WCu_{2.2} ternary catalyst exhibits the lowest PDOS at the Fermi level, suggesting a much lower HBE after alloying with W and Cu. The decreased HBE is beneficial to the HOR.

In the new version of our paper, we have added these data as **Supplementary Figure 37** and provided some discussions therein.

Figure R13. a Calculated PDOS of Ni, Ni₁₇W₃ and Ni_{5.2}WCu_{2.2} with the Fermi level aligned at 0 eV (gray lines). Purple, blue and green dashed line located at the *d*-band center of Ni, Ni₁₇W₃ and

$\text{Ni}_{5.2}\text{WCu}_{2.2}$, respectively. **b Enlarged PDOS with the Fermi level aligned at 0 eV.**

9) *The format of some references is not correct. See Ln. 39, 228, 247.*

Response: We thank the reviewer for pointing out these mistakes and we have revised them accordingly. Moreover, we have read the manuscript very carefully again to ensure the accuracy of our paper.

10) *There are some grammatic errors.*

Response: We thank the reviewer for reading our manuscript carefully and pointing this out to us. We have revised the manuscript very carefully to avoid any grammatical errors that we can find in the new version of our paper.

Reviewer #3 (Remarks to the Author):

Excellent work. The authors present a novel nickel-tungsten-copper catalyst highly active towards HOR in alkaline. The authors claim to achieve a high anode potential up to 0.3 V with a good stability. This works is interesting for the research community. I'd be happy to see this work published after the authors address some minor points:

Response: We thank the reviewer for this very positive assessment of our work and are grateful for his/her recommendation to publish this manuscript in Nature Communications.

1. *Literature refs look not well balanced. Readers should be aware of the ample and intensive activities in the alkaline HOR field. While discussing and reviewing the HOR in alkaline, the authors must cite and discuss recent and VERY relevant studies done on, for instance, Ni-based HOR catalysts for alkaline media done by many other groups (out of my memory, I can think on Atanassov et al., Savinova et al., Dekel et al., Abruna et al., Yan et al., and many others).*

Response: We appreciate the reviewer for pointing out the missing references. This is indeed a negligence on our part. The scientists mentioned by you indeed lead frontier groups on the alkaline HOR research. Following your comments, we have carefully surveyed the works from the listed groups and other groups, and have cited relevant works properly in the revised version of our paper, which we believe could better acknowledge the activity and progress on this intensive research area.

2. *The authors refer to hydroxide exchange membrane fuel cells (HEMFCs). Although this term is used for another 1-2 groups in the world, the common term of >95% of the publications is anion-exchange membrane fuel cells (AEMFCs). The reason for that is that eventually while working with ambient air (containing CO_2), the membrane conducts not only hydroxides but also*

(bi)carbonate anions. At any rate, again, although correct calling it HEMFCs, the most general and correct term may be AEMFCs. Also, the visibility of this term will be wider, which may be of interest for the authors.

Response: We thank the referee for pointing out this potentially unsuitable point and agree to modify the term as AEMFCs. It is true that when working with ambient air (containing CO₂), the membrane conducts not only hydroxides but also (bi)carbonate anions. Thus, to be more precise, we modified the term to AEFMCs throughout the revised manuscript.

3. Authors claim that “... there is no PGM-free HOR catalyst with operating stability 67 up to 0.3 V versus RHE has been reported thus far” – I have found at least one study (ACS Catalysis 9, 6837-6845, 2019; ref 27 in the manuscript) that shows Ni-Fe catalysts with negligible dissolution up to 0.7 V. I had the impression that may be many other similar studies. This is connected to my previous comment re. not a satisfactory literature review. Authors should expand their review of the literature to give a better picture of the field to the readers.

Response: We thank the reviewer for the comments. Indeed, Ref. 27 (ACS Catalysis 2019, 9, 6837-6845) refers to a very impressive experimental study on the chemical and electrochemical stability of Ni₃M (M = Co, Fe, Cu, Mo) electrocatalysts. This work reports that some bimetallic Ni₃M nanoparticles (e.g., Ni₃Fe) show negligible dissolution up to 0.7 V versus RHE, demonstrating remarkable chemical and electrochemical stability. However, these catalysts actually lose their HOR activity at the high anode potentials. For example, as shown in **Figure R14** copied from Ref. 27 (ACS Catalysis 2019, 9, 6837-6845), **all these catalysts lose their HOR currents below 0.2 V versus RHE** (The electrochemical measurements were performed in 0.05 M KOH at a sweep rate of 1 mV s⁻¹ at 25 °C with a rotation of 1600 rpm).

Figure R14. Hydrogen oxidation polarization curves (right panel) for Ni₃Fe/C before (blue) and after (red) various degradation protocols schematically shown in the left panel: 1000 cycles in the potential window of 0–0.3 V_{RHE} (A) and 0–0.7 V_{RHE} (B), as well as before and after 30 days of chemical stability test (C). Conditions for electrochemical measurements: 0.05 M KOH, 25 °C, 1 mV s⁻¹, rotation of 1600 rpm. (This data was copied from *ACS Catalysis* **2019**, 9, 6837-6845).

By comparison, the Ni_{5.2}WCu_{2.2} catalyst reported in this work can stably catalyze HOR in alkali up to 0.3 V versus RHE without the activity loss (see Figure 3a in the revised manuscript). In view of the reviewer’s comments, we also carefully checked the literatures about the alkaline HOR research, and compared the literature results with our data. Again, we can safely state that the HOR stability at an anode potential up to 0.3 V versus RHE has not been achieved by any PGM-free catalysts reported previously.

4. “we speculate that Cu may 88 play an important role in HOR catalysis. Hence, we decide to introduce Cu to form ternary Ni-W-Cu 89 alloy and explore its alkaline HOR property.” – Although fear enough, the readers may want to get a more scientific explanation of why the authors believe copper is the best option.

Response: We thank the reviewer for the comment. While we offered some words to explain why Cu was selected to alloy with Ni and W, we have realized that our discussions might not sufficient. Herein, we would like to clarify this point as following:

First, the oxidation potential of Cu is very positive as compared with other metal elements. For

example, the oxidation of Cu^0 to Cu^{1+} occurs at ~ 0.47 V and the oxidation of $\text{Cu}^0/\text{Cu}^{1+}$ to Cu^{2+} occurs at 0.84 V (*J. Am. Chem. Soc.* **2018**, 140, 16580-16588; *ACS Catal.* **2019**, 9, 6837-6845). Thus, the introduction of Cu into an alloy system would largely improve the structural stability of the alloy catalyst at high anode potentials. Moreover, Dekel and co-workers reported that the behavior of Cu is quite similar to those of the noble metals. For example, the Cu dissolution is of lower concern than Mo dissolution in alloys, which often occurs at a relatively high anode potential that beyond the range of normal fuel cell operation under load (*ACS Catal.* **2019**, 9, 6837-6845).

Second, some previous studies have revealed that the HOR activity and stability in alkaline electrolytes can be enhanced when partially replacing noble metal catalysts with Cu. For examples, Li and co-workers found that BCC-phased PdCu alloy shows superior HOR performance as compared with Pd/C catalyst (*J. Am. Chem. Soc.* **2018**, 140, 16580-16588). Yan and co-workers described that Pt/Cu nanowires competes the HOR properties of Pt nanowires and the state-of-the-art Pt/C catalyst in alkali (*J. Am. Chem. Soc.* **2013**, 135, 13473-13478). These successful attempts also motivate us to consider the alloying of Cu with Ni-W to create HOR catalyst with better stability and stability.

On the basis of above reasons, in our study, we chose the Cu foam as the precursor and substrate to explore new ternary PGM-free alloy catalyst for HOR catalysis in alkaline electrolytes. In the revised manuscript, we have properly modified the relevant discussions to clarify this aspect.

5. Excellent results! The authors compare the performance of Ni, as the base case, with NiW, and NiWCu. Although this is very informative, the stoichs used are completely different. I find it difficult to understand the comparison between Ni_5WCu_2 and Ni_{17}W_3 . Somehow a better comparison will be adding $\text{Ni}_{17}\text{W}_3\text{Cu}_x$ and Ni_5W , for instance. Also, why these specific compositions were selected is not clear.

Response: We appreciate the reviewer for the thoughtful criticism. We understand the reviewer's inquiry about the stoichiometric ratio of the alloy catalysts. Well, as illustrated in **Figure 1** in our original manuscript, we used the bulky Cu foam as the precursor, which can support the growth of Cu nanowires throughout the foam. After that, we treated the resultant foam with $\text{Ni}(\text{NO}_3)_2 \cdot 6\text{H}_2\text{O}$, $(\text{NH}_4)_6\text{H}_2\text{W}_{12}\text{O}_{40} \cdot x\text{H}_2\text{O}$, and $\text{CO}(\text{NH}_2)_2$ in deionized water at 130 °C for 8 h in a hydrothermal system, which can yield the NiW-Cu(OH)₂ precursor. We eventually achieved the $\text{Ni}_{5.2}\text{WCu}_{2.2}$ alloy by annealing the NiW-Cu(OH)₂ precursor in H_2/Ar (5/95) atmosphere at 500 °C for 1 h. Although this strategy enables us to obtain new Ni-W-Cu ternary alloy as active and stable HOR catalyst, we also want to state that we are not able to precisely control the stoichiometry of the alloys, presumably owing to the multi-step process.

However, this strategy can allow us to **roughly tune the stoichiometry** by changing the ratio of the

$\text{Ni}(\text{NO}_3)_2 \cdot 6\text{H}_2\text{O}$ and $(\text{NH}_4)_6\text{H}_2\text{W}_{12}\text{O}_{40} \cdot x\text{H}_2\text{O}$ during the synthesis. As shown in **Figure R15** below, we can synthesize Ni-W-Cu alloys with stoichiometric ratios of $\text{Ni}_{3.9}\text{WCu}_{1.5}$, $\text{Ni}_{5.7}\text{WCu}_{1.8}$, $\text{Ni}_{5.2}\text{WCu}_{2.2}$, and $\text{Ni}_9\text{WCu}_{1.8}$ by turning the ratio of the $\text{Ni}(\text{NO}_3)_2 \cdot 6\text{H}_2\text{O}$ and $(\text{NH}_4)_6\text{H}_2\text{W}_{12}\text{O}_{40} \cdot x\text{H}_2\text{O}$. Electrochemical measurements reveal that the $\text{Ni}_{5.2}\text{WCu}_{2.2}$ alloy shows superior HOR activity as compared with Ni-W-Cu alloys with other stoichiometric ratios (**Figure R15**). We thus selected the $\text{Ni}_{5.2}\text{WCu}_{2.2}$ as the representative Ni-W-Cu alloy for discussion.

In our original manuscript, we compared the HOR performance of $\text{Ni}_{5.2}\text{WCu}_{2.2}$ with that of Ni and Ni_{17}W_3 . The aim of such comparison is to reveal the important role of Cu, which alloys with Ni and W to form Ni-W-Cu ternary alloy, exhibiting substantially improved HOR performance as compared to the binary Ni_{17}W_3 alloy. However, as described above, we are also unable to precisely control the stoichiometry of Ni-W alloy. For example, we can not synthesize the $\text{Ni}_{5.2}\text{W}$ alloy as suggested by the reviewer, partially because that there is no such $\text{Ni}_{5.2}\text{W}$ phase exists.

Overall, we note that the precise stoichiometric control of the alloys is challenging at the present time. Currently, their stoichiometry can be roughly controlled. In the new version of our paper, we added the experimental data about Ni-W-Cu ternary alloy with different stoichiometric ratios as **Supplementary Figure 20**, and provided some discussion therein.

Figure R15. Material characterization and HOR activities of Ni-W-Cu alloys with different Ni/W/Cu ratios. a-h, SEM and TEM images of catalysts obtained under different addition amounts of Ni and W precursors: Ni_{3.9}WCu_{1.5} (**a, e**), Ni_{5.7}WCu_{1.8} (**b, f**), Ni_{5.2}WCu_{2.2} (**c, g**), Ni₉WCu_{1.8} (**d, h**). **i**, XRD patterns. **j**, HOR polarization curves. **k**, EIS Nyquist plots. The catalyst with Ni/WCu ratio of 5.2:1:2.2 shows the best HOR performance. All the samples are obtained by annealing at 500 °C for 1.0 h.

6. *At which temperature the Chronoamperometry responses were carried out in fig 3?*

Response: The Chronoamperometry experiment was performed at room temperature (around 25 °C). We have clarified this in the revised version of our paper.

7. *Although HBE is critical, some other studies indicate that OHBE may be also used as indicator for HOR in alkaline (see for instance, Catalysts, 8(10), 454, 2018). Readers may be very interested in this discussion.*

Response: Thanks for the comments. We read very carefully the paper (*Catalysts*, **2018**, 8(10), 454) mentioned by the reviewer and found that it refers to an interesting study on the effect of metal doping on the activity of Ni-based HOR electrocatalysts. In that work, both HER and OHBE were thought to be important parameters to describe the HOR activity of Ni-based catalysts in alkali. As a matter of fact, in our original manuscript, we also demonstrate that this bifunctional mechanism works on the Ni_{5.2}WCu_{2.2} alloy catalyst based on our experimental and computational results (see pages 11-12 in the main text).

In view of the reviewer's comments, we have made some modifications to the relevant discussions, which we believe could better clarify the bifunctional mechanism. Moreover, the missing original reference (*Catalysts*, **2018**, 8(10), 454) has been included as the Ref. 50 in our revised manuscript.

8. *Although the results are very clear and informative, the science behind is not clear. Why this approach does improve the stability of the Ni catalyst to a potential 0.3 V?*

Response: We thank the reviewer for the comment and question, which are similar to the reviewer's 4# comments. As we discussed above, the oxidation potential of Cu is very positive as compared to other metal elements. For example, the oxidation of Cu⁰ to Cu¹⁺ occurs at ~0.47 V and the oxidation of Cu⁰/Cu¹⁺ to Cu²⁺ occurs at 0.84 V (*J. Am. Chem. Soc.* **2018**, 140, 16580-16588; *ACS Catal.* **2019**, 9, 6837-6845). Thus, alloying Cu with Ni and W would largely improve the structural stability of the resultant alloy catalyst at high anode potentials. Additionally, Dekel and co-workers demonstrated that the behavior of Cu is quite similar to those of the noble metals. For example, the Cu dissolution largely suppressed in alloys as compared to other metals such as Mo, which often happens at a high anode potential that beyond the range of normal fuel cell operation under load (*ACS Catal.* **2019**, 9, 6837-6845).

We thus conclude that the above merits of Cu lead to the improved HOR stability up to 0.3 V versus RHE. In the new version of our paper, we have properly modified the relevant discussions to clarify this point.

9. *It will be of interest to show the oxygen map in Figure 2c.*

Response: Thanks for the nice suggestion. Actually, we performed the EDX spectrum elemental mapping on oxygen. As shown in **Figure R16** below, the oxygen signal is much weaker as compared with Ni, W, and Cu, which could be the adsorbed oxygen on the surface of Ni_{5.2}WCu_{2.2} alloy when exposing the sample in air. Further, our XRD pattern also supports the formation of alloy phase (see **Figure 2d** in the revised manuscript). No any metal oxides were detected. Considering that the oxygen signal comes from surface adsorbed oxygen, we thus did not show the oxygen map in our original manuscript.

Figure R16. STEM-EDX elemental mappings. It shows that O signal is relatively weaker than Ni, W, and Cu, which could be from the adsorbed O and slight surface oxidation when exposing the sample in the air.

Following your suggestion, we now added the oxygen map as **Supplementary Figure 8** in the revised Supplementary Information and provided some discussion therein.

10. *Why the limiting current for the HOR in Figure 3a,e,f is too high compared to the theoretical value? What is the rotation rate used for HOR? Please check articles for theoretical limiting (among many others, <https://doi.org/10.1002/adfm.202002087>)*

Response: We thank the reviewer for this very thoughtful and excellent questions! Indeed, we achieved higher limiting currents than the theoretical value on both Ni_{5.2}WCu_{2.2} monolith and Pt/C-modified Cu foam. We note that these HOR curves are highly repeatable and thus are correct. We elucidate the underlying reasons as follow:

(1) As to the RDE test, the HOR catalysts tend to have the same diffusion limiting currents (theoretical value) at a fixed rotating speed according to the Levich equation (*Analytical Chemistry*,

1962, 34, 164): $i_l = 0.62nFAD^{2/3}\nu^{-1/6}c_0\omega^{1/2} = ABc_0\omega^{1/2}$, where i_l is the limiting current, D is the diffusivity of hydrogen in electrolyte, n is the number of electrons transferred in HOR ($n = 2$), A is the area of the electrode, ν is the kinematic viscosity of the electrolyte and c_0 is the solubility of H_2 in electrolyte, ω is the rotation rate and $B = 0.62nFD^{2/3}\nu^{-1/6}$. The equation reveals that the value of i_l depends solely on the rotation speed ω . **It should be noted that the Levich equation can not be applied when the value of ω is small. This is because that a small ω would cause the hydrodynamic boundary layer becomes very large as compared to the disk radius, thus the assumption used in derivation of the Levich equation no longer holds (*J. Electrochem. Soc.* 2015, 162, F1470-F1481).**

(2) As to the $Ni_{5.2}WCu_{2.2}$ catalyst, our current synthetic strategy can produce the bulky $Ni_{5.2}WCu_{2.2}$ monolith (see **Figure 1** in the revised manuscript). Thus, we directly used the $Ni_{5.2}WCu_{2.2}$ monolith and the Pt/C-modified Cu foam as electrodes for HOR studies. **That is to say, no rotation can be applied to these bulky electrodes (thus $\omega = 0$), leading to the inapplicability of the Levich equation here.** In comparison to the catalyst on the RDE electrode, our bulky $Ni_{5.2}WCu_{2.2}$ monolithic electrode without rotation is highly porous, which enables the exposure of much more catalytic active sites, thus leading to the observed currents.

On the basis of the reviewer's question, we have added some discussions in our revised manuscript to clarify this aspect, where the interesting work mentioned by the reviewer was included to discuss.

11. Typically, Ni-based catalysts are poised to a greater extent with CO_2 than CO . Therefore, it will be interesting and more helpful for the alkaline environment, where CO_2 cannot be avoided (see for instance <https://doi.org/10.1021/acscatal.6b00487>)

Response: We appreciate the reviewer for the thoughtful comments. The reviewer referred to a very interesting experimental study regarding electrochemical oxidation of urea on nanostructured $LaNiO_3$ perovskite (*ACS Catal.* 2016, 6, 5044-5051). In that work, Stevenson *et al.* reported that CO_2 could be a catalyst poison that leads to the deactivation of $LaNiO_3$ perovskite catalyst, which potentially caused by the formation of nickel carbonate on the catalyst surface.

Motivated by the reviewer's comments, we also compared the HOR activity of our $Ni_{5.2}WCu_{2.2}$ alloy catalyst in pure H_2 and in H_2 with 20,000 ppm CO_2 . Indeed, our electrochemical measurements show that the HOR activity of $Ni_{5.2}WCu_{2.2}$ alloy decreases with 20,000 ppm CO_2 impurity in the H_2 fuel (**Figure R17** below). Interestingly, we further found that the HOR activity of Pt/C catalyst also decreased when 20,000 ppm CO_2 exists in H_2 (**Figure R17** below). On the basis of these results, we conclude that, although $Ni_{5.2}WCu_{2.2}$ catalyst could be poisoned by CO_2 , which might be not caused by the formation of nickel carbonate on the catalyst surface like $LaNiO_3$ perovskite. The detailed poisoning mechanism is rather complex and needs future investigations.

Figure R17. HOR polarization curves for Ni_{5.2}WCu_{2.2} alloy and Pt/C in H₂-saturated 0.1 M KOH with (dashed lines) and without (solid lines) the presence of 20,000 ppm CO₂.

In our revised manuscript, we added the new results in the Supplementary Information and provided some discussions therein.

12. In Figure 3(c, d), the author may want to normalize the j_0 with the electrochemical surface to get the intrinsic activity to avoid the effect of loading and particle size effect.

Response: We thank the reviewer for the useful suggestion. We know that the best way to evaluate the intrinsic electrochemical activity of a catalyst is to calculate its specific activity based on its electrochemically active surface area (ECSA). However, the ECSA values are difficult to obtain for many non-noble metals: they cannot be calculated using the classic hydrogen under-potential deposition (UPD) like commonly done for Pt because no obvious hydrogen adsorption occurs prior to H₂ evolution. Moreover, the CO stripping method for calculating the ECSA is commonly applicable to some noble metals (Pt, Pd, etc.). Therefore, as to the non-noble catalysts, we have to turn to other methods to evaluate the ECSAs.

Because our synthetic strategy gave rise to the large Ni_{5.2}WCu_{2.2} monolith (see **Figure 1** in the revised manuscript). We thus turn to find a method that could be used to determine the ECSA of such bulky catalytic materials. Previous studies reveal that the ECSA values of monolithic catalysts can be determined from the double-layer capacitance according to equation $ECSA = C_{dl}/C_s$ (where C_s is the specific capacitance of the catalyst or the capacitance of smooth planar surface per unit area) that suggested by McCrory, Peters and Jaramillo (*J. Am. Chem. Soc.* **2013**, 135, 16977). For examples, Sun and co-workers used this method to calculate the ECSA of bulky Ni₃Ni/Ni/Ni-foam catalyst

(*Nat. Commun.* **2018**, 9, 4531-4540) as well as Co₂N/Co/Cobalt-foam catalyst (*ACS Energy Lett.* **2019**, 4, 1594-1601).

Thus, we calculated the ECSA values of our studied catalysts from the double-layer capacitance according to equation $ECSA = C_{dl}/C_s$. For the estimate of surface area, we adopted the general specific capacitance of $C_s = 11 \mu\text{F cm}^{-2}$ based on typical reported value (*Angew. Chem. Int. Ed.* **2019**, 58, 10644-10649). As shown in **Figure R18** below, our measurements give ECSA values of 818 cm² for Ni_{5.2}WCu_{2.2} monolith, 173 cm² for Ni₁₇W₃, 64 cm² for Ni, and 1582 cm² for Pt/C. On the basis of the more accurate ECSA values, we thus could offer a fair comparison of our studied catalysts that shown in **Figure R18f**.

In the new version of our paper, we replaced the BET-normalized j_0 with the ECSA-normalized one (see **Figure 3c** in the revised manuscript) and offered some discussions over there. Regarding the **Figure 3d** in our original manuscript, all its curves were derived from the **Figure 3a** to obtain the HOR/HER Tafel plots, thus its j_k (kinetic current density) was normalized to the geometric surface area.

Figure R18. Electrochemically active surface area (ECSA) measurements. CV curves of **a** $\text{Ni}_{5.2}\text{WCu}_{2.2}$, **b** Ni_{17}W_3 , **c** Ni, and **d** Pt/C collected at various scan rates ranging from 4 to 20 mV s^{-1} in CH_3CN with 0.15 M KPF_6 . **e** The linear fitting of scan rate versus Δj (the difference between the anodic and cathodic current densities at open circuit potential). **f**. Comparison of exchange current density (j_0) of various studied catalysts normalized by geometric areas (unpatterned) and ECSA (patterned), respectively.

13. Including data of AEMFC test with the NiWCu best catalyst will be of great value.

Response: We thank the reviewer for the thoughtful suggestion and fully agree on that including the AEMFC data would further strengthen the paper. We note that the $\text{Ni}_{5.2}\text{WCu}_{2.2}$ alloy was made from bulky Cu foam, thus only macroscopic target catalyst can be synthesized at the present time. Such

bulky catalyst can not be directly used as gas diffusion electrode for MEA measurements. To surmount this challenge, we are now exploring methods to synthesize high-quality $\text{Ni}_{5.2}\text{WCu}_{2.2}$ powder catalyst, which can be deposited onto a carbon-based gas diffusion layer for MEA study. Therefore, we highlight that although the AEMFC testing is a significant undertaking and may not be possible within the scope of the present work, we will report related results later once the $\text{Ni}_{5.2}\text{WCu}_{2.2}$ powder is attainable.

REVIEWERS' COMMENTS

Reviewer #1 (Remarks to the Author):

The reviewers' comments are well responded. I recommend this article to be published on Nature Communications.

Reviewer #2 (Remarks to the Author):

The authors have successfully addressed all the issues I raised. I, therefore, recommend this paper for publication.

Reviewer #3 (Remarks to the Author):

The authors addressed all my comments in a very good way. I am happy to recommend the manuscript for publication.